# Localizations of Laminin Chains Suggest Their Multifaceted Functions in Mouse Tooth Development

**DOI:** 10.3390/ijms26094134

**Published:** 2025-04-26

**Authors:** Tian Liang, Hong Zhang, Yuanyuan Hu, Mansi Solanki, Chuhua Zhang, Takako Sasaki, Charles E. Smith, Jan C.-C. Hu, James P. Simmer

**Affiliations:** 1Department of Orthodontics and Pediatric Dentistry, University of Michigan School of Dentistry, 1011 North University, Ann Arbor, MI 48109, USA; yyhu@umich.edu (Y.H.); mansisol@umich.edu (M.S.); 2Department of Biologic and Materials Sciences, University of Michigan School of Dentistry, 1011 North University, Ann Arbor, MI 48109, USA; zhanghon@umich.edu (H.Z.); chuhuaz@umich.edu (C.Z.); charles.smith@mcgill.ca (C.E.S.); janhu@umich.edu (J.C.-C.H.); jsimmer@umich.edu (J.P.S.); 3Department of Matrix Medicine, Faculty of Medicine, Oita University, 1-1 Idaigaoka, Hasama-Machi, Yufu City 879-5593, Oita, Japan; tsasaki@oita-u.ac.jp; 4Department of Anatomy & Cell Biology, Faculty of Medicine & Health Sciences, McGill University, 3640 University St., Montreal, QC H3A 0C7, Canada

**Keywords:** laminin, basement membrane, Tomes’ process, ameloblast, enamel, tooth development

## Abstract

The human laminin family is composed of five α, four β, and three γ chains. Laminins are heterotrimers of α, β, and γ chains. Laminins play critical roles during organogenesis, mostly as basement membrane components. The expression of all and the localization of most laminin chains were characterized in mouse developing teeth. Primary laminin isoforms in basement membranes along the inner enamel epithelium before the secretory stage and outside of the outer enamel epithelium were laminins 111 (α1β1γ1) and 511. The mouse laminin α3 chain has two variants, α3A and α3B. Although a basement membrane structure is absent, laminin 3A32 was localized along the secretory surface of the secretory stage ameloblast Tomes’ processes. Laminin 3A32 was localized along the atypical basement membrane of maturation stage ameloblasts and the specialized basement membrane of junctional epithelium facing the enamel surface. The endothelial basement membrane in the dental papilla and outside of the enamel organ contained laminins 411 and 511. Laminin 332 was detected in the extracellular matrix but not the basement membrane of the apical loop. Laminin 111 was localized in the extracellular matrix of the apical dental papilla without forming a visible basement membrane. These findings suggest the multifaceted functions of laminins in tooth development and set the foundation for functional investigations.

## 1. Introduction

A basement membrane, or basal lamina, is a thin, dense sheet of membrane-bound extracellular matrix at the interface between most epithelium and endothelium and neighboring tissues [1] with a dynamic and distinctive composition appropriate for each tissue’s unique needs. The basement membrane prototype contains laminin, collagen IV, nidogen, and perlecan [2]. Laminin and collagen IV trimeric proteins independently assemble into networks, which are interconnected with nidogen, perlecan, and other molecules [2,3]. Under electron microscopy, basement membranes appear as three layers: lamina lucida (electron-lucent), lamina densa (electron-dense), and lamina reticularis (electron-lucent) [4,5]. Basal lamina, sometimes used interchangeably with the basement membrane, includes lamina lucida and lamina densa, but not lamina reticularis [5]. Initially regarded as a physical barrier to separate tissues of distinct developmental origins and to selectively restrict molecular permeability of the cell membrane, it is now appreciated that the basement membrane is an active component regulating cell behaviors [2,6,7]. Basement membranes establish cellular polarity and regulate cell adhesion, migration, tissue morphogenesis, and signaling [6]. Disruption of basement membranes can lead to lethal developmental defects and cancer metastasis [6].

Laminins that form a fundamental network of basement membranes are large heterotrimers consisting of α, β, and γ chains. In humans, there are five α, four β, and three γ chains, while the β4 chain is missing in rodents [8]. Taking the two α3 chain variants α3A and α3B into account, there are at least 16 laminin isoforms identified in tissues [9]. All laminin isoforms share a common structure: three short arms consisting of the N termini of three individual chains extend from a coiled-coil long arm consisting of all three chains, followed by the C terminus of the α chain [6,8]. The short arms are critical for laminin polymerization into the planar network, while the C terminus of the α chain can bind to molecules like integrins to transduce signaling to adjacent cells [3,8]. Structural differences and tissue-specific expression of laminin isoforms contribute to the functional diversity of basement membranes.

Tooth germs develop from a series of reciprocal interactions between the oral epithelium and mesenchyme [10]. The epithelial component of the tooth germ forms the enamel organ. It consists of outer enamel epithelium (OEE), stellate reticulum (SR), stratum intermedium (SI), and inner enamel epithelium (IEE) that differentiate into ameloblasts to direct enamel formation and mineralization [11]. Basement membrane structures exist at the interface between dental epithelium and dental mesenchyme, specifically between OEE and dental follicle and between IEE and dental papilla. The basement membrane along the distal aspect of differentiating ameloblasts is degraded and resorbed prior to the secretory stage of enamel formation [12]. Soon after the onset of dentin mineralization and islands of dentin coalescing into a continuous mineral layer, initial enamel mineral ribbons form on the surface of mineralized dentin and the secretory surface of the ameloblast distal membrane [13]. The initial elongation of enamel ribbons allows the formation of Tomes’ processes at the distal end of secretory stage ameloblasts [14], which leads to the rod-interrod enamel architecture [15]. Once the thickness of enamel is established, ameloblasts transition into the maturation stage, where an atypical basement membrane is formed at their distal end [16]. During the maturation stage, the enamel mineral deposition is rapidly increased to achieve ~96% mineral content by weight, making the dental enamel the hardest tissue in humans [16].

Genetic defects in laminin genes have significant consequences in tooth development and enamel formation. The dental epithelium of *Lama5* (which encodes the laminin α5 chain) knockout mice fails to proliferate and differentiate [17]. Biallelic mutations in *LAMA3*, *LAMB3*, and *LAMC2* cause junctional epidermolysis bullosa with severe enamel hypoplasia in humans [18,19,20,21], while single allelic defects in *LAMA3* [22,23,24] and *LAMB3* [25,26,27,28,29] cause localized enamel defects without skin fragility. Mice without *Lama3* show neonatal lethality and abnormal ameloblast morphology and differentiation [30].

Previously, all five laminin α chains were found to be expressed during early molar morphogenesis [31]. LAMA1 and LAMA5 were localized along the basement membrane separating the dental epithelium and mesenchyme, while the two *Lama3* variants appeared to be differentially expressed by cells in enamel organ epithelium [31,32]. There was controversy about whether laminin 332 (a heterotrimer of laminin α3, β3, and γ2 chains) was in the basement membrane separating dental epithelium and mesenchyme. Laminin 332 was localized along the presecretory basement membrane [33]. However, *Lama3*, *Lamb3*, and *Lamc2* transcripts were detected in secretory ameloblasts [34,35,36], although a basement membrane structure was absent. Laminin 332 was localized along the atypical basement membrane along the distal surface of maturation ameloblasts [37] and the specialized basement membrane between the junctional epithelium and the matured enamel surface [38]. Particularly, the *Lama3* variants that were expressed by ameloblasts during enamel mineralization were largely unclear. Considering the structural diversity of laminin chains (even between α3A and α3B chains) and the functional diversity of laminin isoforms, it is critical to comprehensively characterize the expression of all laminin genes and the localization of all laminin isoforms in developing teeth. This will lay the foundations for us to decipher the molecular mechanisms of dental, particularly enamel, defects associated with laminins.

In this study, RNAscope *in situ* hybridization and immunohistochemistry are used to elucidate the expression and localization of a complete panel of individual laminin chains in mouse developing incisors and molars with the objective to provide a foundation for subsequent functional studies. We hypothesize that laminin chains are differentially expressed in developing teeth and that multiple laminin isoforms are differentially distributed in the extracellular matrix of developing teeth.

## 2. Results

RNAscope *in situ* hybridization was performed using a complete set of riboprobes specifically targeting 5 α, 3 β, and 3 γ laminin chains, as well as *Col4a1* and *Col7a1*, in developing mouse teeth, including maxillary molars at different developmental stages and continuously growing mandibular incisors (Appendix A). Immunohistochemistry was performed using available and validated antibodies [39,40] against type IV collagen and all laminin chains, except for LAMB2 and LAMC3. Results were presented in Appendix A.

Here, our findings were presented with three major focuses: laminins associated with the enamel organ epithelium and its derivative (Section 2.1, Section 2.2, Section 2.3 and Section 2.4 and Figure 1, Figure 2, Figure 3, Figure 4, Figure 5, Figure 6, Figure 7 and Figure 8), the basement membrane of vascular supplies to tooth germs (Section 2.5 and Figure 9), and laminin components in the extracellular matrix of apical dental papilla cells (Section 2.6 and Figure 10). Regarding laminins associated with the enamel organ epithelium and its derivative, findings on basement membrane structures that were not directly associated with mineralization (Section 2.1), laminins associated with secretory (Section 2.2) and maturation (Section 2.3) stages ameloblasts, and laminins in the junctional epithelium (Section 2.4) were presented. Seminal findings of RNAscope and immunohistochemistry from incisors were present in Figure 1, Figure 2 and Figure 3, while findings of RNAscope and immunohistochemistry from molars of different stages were present in Figure 4, Figure 5 and Figure 6. RT-PCR results to distinguish the *Lama3* transcript were shown in Figure 7. Immunohistochemical results on the junctional epithelium were shown in Figure 8.

### 2.1. Primary Laminin Isoforms in the Basement Membrane Structures Surrounding Inner and Outer Enamel Epithelium Were Laminins 511 and 111

Prior to the onset of tooth mineralization, basement membrane structures separate the dental epithelium and dental mesenchyme at the interface both between the IEE and pre-odontoblasts (a part of the dental papilla) and between the OEE and the dental follicle. In continuously growing mouse incisors that contain all stages of tooth development in a single longitudinal section, the enamel organ epithelium expressed *Col4a1* (Figure 1), which encodes an α chain of type IV collagen. As shown by immunolabeling, the basement membranes contained type IV collagen (Figure 2 and Figure 3), a principal basement membrane component. The basement membrane between the distal membrane of the IEE and pre-odontoblasts appeared relatively linear under the light microscope (Figure 2 and Figure 3). The basement membrane between the OEE and the dental follicle was relatively straight at first (Figure 2 and Figure 3). When the corresponding IEE differentiated into secretory stage ameloblasts with Tomes’ processes, the basement membrane adjacent to the OEE appeared tortuous as the outer surface of the enamel organ started to invaginate (Figure 2 and Figure 3). During the maturation stage of enamel formation, SI, SR, and OEE condensed into a papillary layer. The outer surface of the papillary layer that contained type IV collagen was even more tortuous (Figure 2 and Figure 3). This basement membrane loop structure was one cell layer apart from the proximal end of maturation stage ameloblasts intermittently. *Col7a1* encodes the α chain of type VII collagen that forms the anchoring fibrils of lamina reticularis in the basement membrane. *Col7a1* was expressed by IEE and OEE cells adjacent to these two basement membrane structures (Figure 1).

Other than types IV and VII collagen, basement membrane structures between the IEE and dental papilla and between the OEE and dental follicle contained multiple laminin isoforms. The mRNA of *Lama5*, *Lamb3*, and *Lamc1* were moderately detected in differentiating IEE, where the mRNA of *Lama1* and *Lamb1* were detected weakly (Figure 1). Immunolabeling of these two basement membrane structures clearly showed that they contained LAMA1, LAMA5, LAMB1, and LAMC1, indicating that these two basement membrane structures contain laminin isoforms 111 and 511 (a heterotrimer of laminin α1, β1, and γ1 chains and a heterotrimer of laminin α5, β1, and γ1 chains, respectively). Although the basement membrane between the IEE and dental papilla degrades following IEE differentiation into secretory stage ameloblasts, the basement membrane along the outer surface of OEE and the papillary layer remained intact under light microscopy (Figure 2A).

*Lama3*, *Lamb3*, and *Lamc2* expression was detected in the labial apical loop (Figure 1). The extracellular matrix of enamel organ apical loop epithelial cells was positive for LAMB3, LAMC2, and laminin 332 (Figure 2), without these signals forming a planar pattern. The apical loop was not immunolabeled by an antibody specifically targeting the LAMA3A isoform domain IIIa [40,41] (Figure 2, Appendix A), suggesting that the LAMA3 isoform in the apical loop extracellular matrix is LAMA3B. The basement membrane between the OEE and the dental follicle was intermittently and vaguely positive for laminin 332 (Figure 2 and Figure 3), suggesting that laminin 332 is a minor component of this basement membrane structure.

Newborn, 4-day-old, and 10/12-day-old developing mouse first molars were selected to reveal the laminin chain expression patterns in the presecretory, secretory, and maturation stages of enamel organ epithelial cells. Similar to our findings in mouse incisors, strong expression of *Lama5* and *Lamb3* and weak expression of *Lama1*, *Lamb1*, and *Lamc1* were observed in the newborn presecretory IEE (Figure 4). The cervical loop of 4-day-old maxillary first molars was wrapped by a basement membrane structure that clearly contained type IV collagen, LAMA1, LAMA5, LAMB1, and LAMC1 (Figure 5). Later during tooth development, the cervical loop forms Hertwig’s epithelial root sheath (HERS), which is critical for tooth root development [42]. Before its disintegration, the epithelium-derived HERS was separated from the mesenchyme-derived dental papilla (on the inside) and the dental follicle (on the outside) by basement membrane structures containing type IV collagen, LAMA1, LAMA5, LAMB1, and LAMC1 (Figure 6). The basement membrane structure outside the OEE during the secretory stage (Figure 5) and the papillary layer during the maturation stage (Figure 6) also contained type IV collagen, LAMA1, LAMA5, LAMB1, and LAMC1, while the contour of this basement membrane structure becomes more tortuous as enamel mineralization progresses.

**Figure 1 ijms-26-04134-f001:**
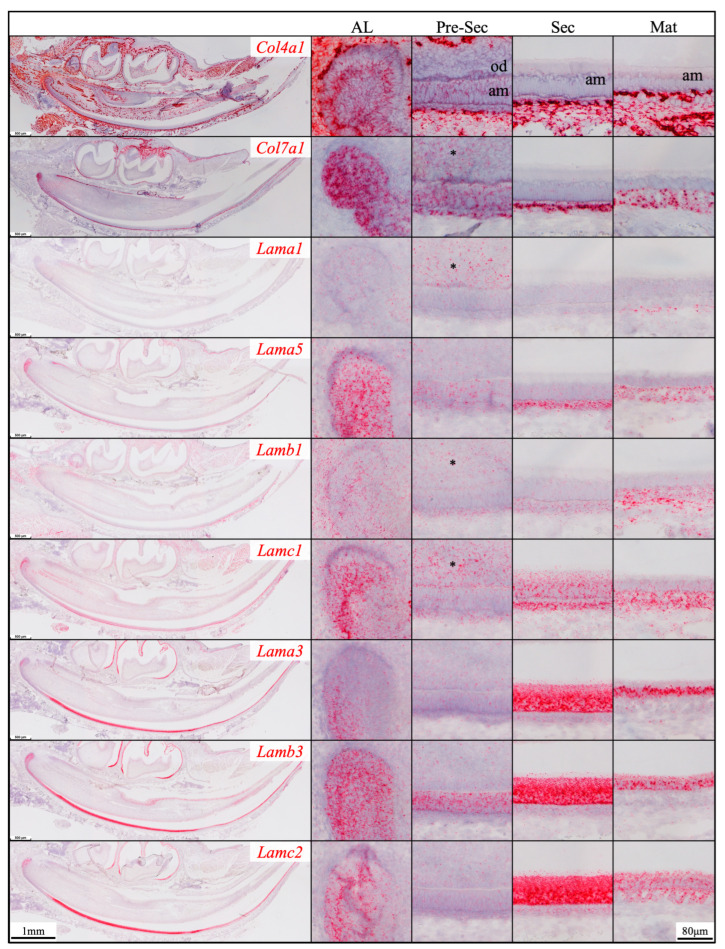
RNAscope *in situ* hybridization of *Col4a1*, *Col7a1*, and laminin chains in 10-day-old mouse mandibular incisors. mRNA signals are in red. Mouse continuously growing incisors contain ameloblasts (am) of all developmental stages. Dental epithelial stem cells reside in the apical loop (AL). The inner enamel epithelial cells during the pre-secretory (Pre-Sec) stage actively proliferate and then differentiate into secretory (Sec) stage ameloblasts that guide the appositional growth of enamel ribbons. Enamel is further hardened during the maturation (Mat) stage. The enamel organ epithelium at the apical loop expresses *Col4a1*, *Col7a1*, and laminin α1, α5, β1, γ1, α3, β3, and γ2 chains. Pre-secretory stage ameloblasts express *Col4a1*, *Col7a1*, and laminin α5 and β3 chains. Secretory and maturation stage ameloblasts express laminin γ1, α3, β3, and γ2 chains. Adjacent to odontoblasts (od), the apical dental papilla cells (*) express *Col7a1*, *Lama1*, *Lamb1*, and *Lamc1*.

**Figure 2 ijms-26-04134-f002:**
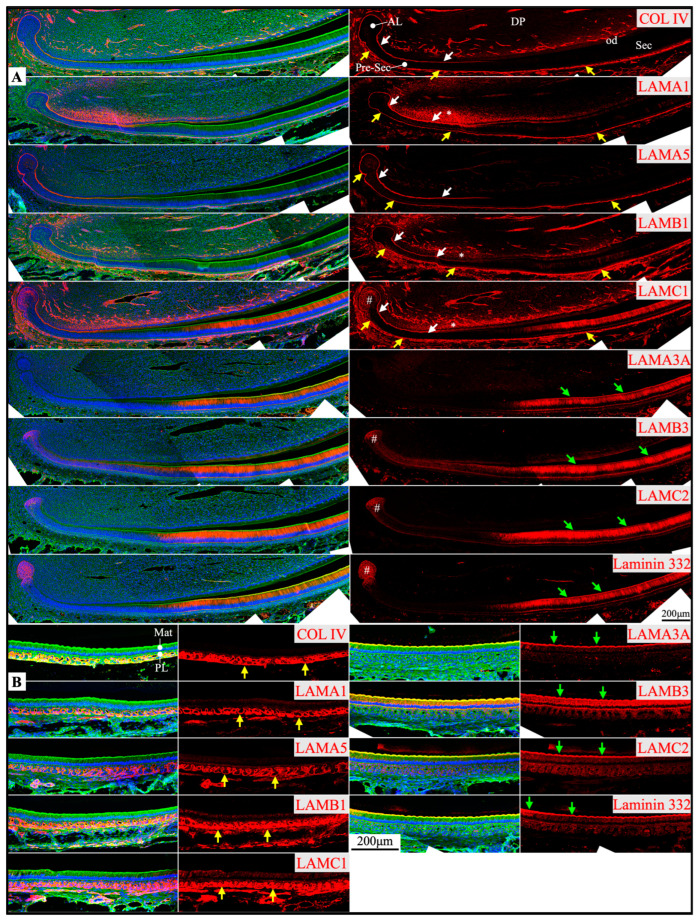
Immunohistochemistry of type IV collagen and laminin chains in 10-day-old mouse mandibular incisors. Type IV collagen (COL IV) and laminin chains are red. The cytoskeleton labeled by β-actin is green. Nuclei labeled by DAPI are blue. Mouse continuously growing incisors contain ameloblasts of all developmental stages, from the dental epithelial stem cells in the apical loop (AL) to pre-secretory (Pre-Sec) and secretory (Sec) stage ameloblasts (**A**) and further into maturation (Mat) stage ameloblasts (**B**). The enamel organ during the maturation stage, except for the ameloblasts, condenses into a papillary layer (PL). The basement membrane between the inner enamel epithelium and dental papilla (DP) (white arrows) contains COL IV, LAMA1, LAMA5, LAMB1, and LAMC1. This basement membrane structure is degraded before the onset of the secretory stage. The basement membrane between the outer enamel epithelium and dental follicle (yellow arrows) also contains COL IV, LAMA1, LAMA5, LAMB1, and LAMC1. The distal aspects of secretory and maturation stage ameloblasts (green arrows) are labeled by LAMA3A, LAMB3, LAMC2, and laminin 332. The extracellular matrix of the enamel organ apical loop (#) is positive for LAMC1, LAMB3, LAMC2, and laminin 332. The extracellular matrix of apical dental papilla cells (*) adjacent to odontoblasts (od) is positive for LAMA1, LAMB1, and LAMC1.

**Figure 3 ijms-26-04134-f003:**
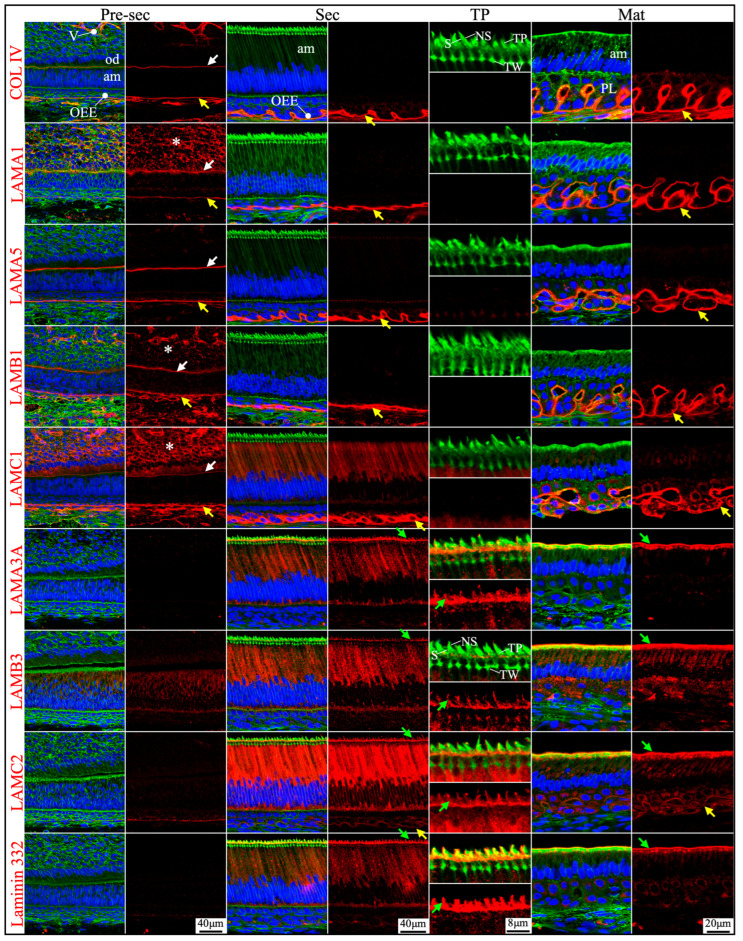
Immunohistochemistry of type IV collagen and laminin chains in 10-day-old mouse mandibular incisors. Lower magnification images of mandibular incisors shown are in Figure 2. The basement membrane between presecretory (Pre-Sec) stage ameloblasts (am) and odontoblasts (od) (white arrows) comprises type IV collagen (COL IV), LAMA1, LAMA5, LAMB1, and LAMC1. The basement membrane immediately outside of the outer enamel epithelium (OEE) (yellow arrows) also contains type IV collagen, LAMA1, LAMA5, LAMB1, LAMC1, and a low level of LAMC2, and is becoming more tortuous as enamel mineralization progresses. In comparison, the distal aspects of secretory (Sec) and maturation (Mat) stage ameloblasts (green arrows) are positive for LAMA3A, LAMB3, LAMC2, and laminin 332. Particularly, the secretory surface (S) (green arrows) but not the non-secretory surface (NS) of the Tomes’ processes (TP) is positive for LAMA3A, LAMB3, LAMC2, and laminin 332. LAMC1 is detected in the supranuclear region of secretory stage ameloblasts, but the signal is not detected beyond the distal terminal webs (TW). The extracellular matrix of apical dental papilla cells (*) adjacent to odontoblasts (od) is positive for LAMA1, LAMB1, and LAMC1. Key: V, blood vessels.

**Figure 4 ijms-26-04134-f004:**
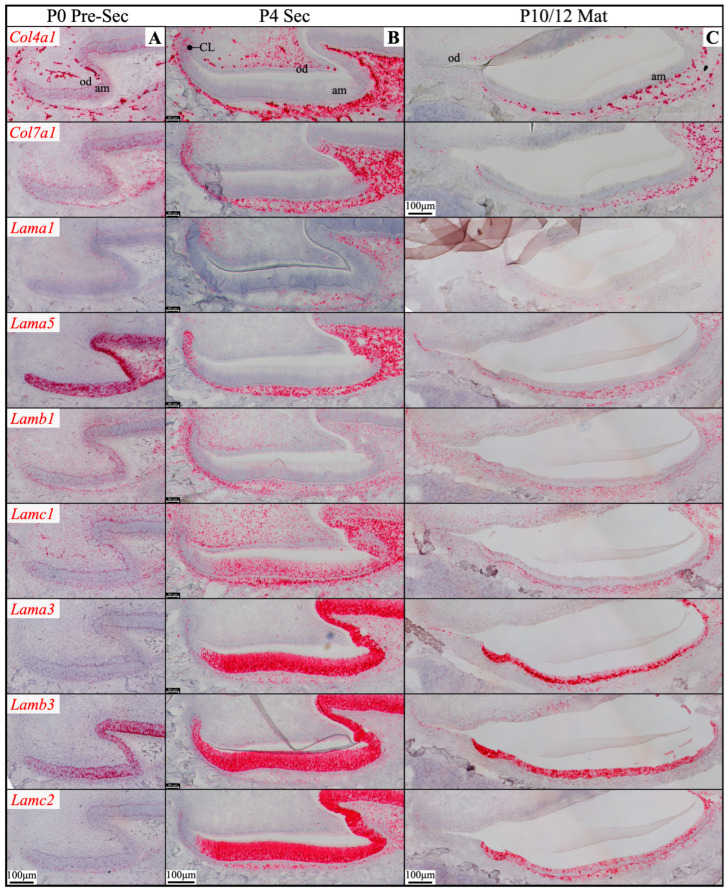
RNAscope *in situ* hybridization of *Col4a1*, *Col7a1*, and laminin chains in mouse maxillary 1st molars. Ameloblasts (am) in the P0 (newborn, **A**), P4 (4-day-old, **B**), and P10/12 (10/12-day-old, **C**) enamel organ epithelium are mostly in pre-secretory (Pre-Sec), secretory (Sec), and maturation (Mat) stages, respectively. For (**C**), only sections for *Col4a1* and *Col7a1* are from P10 samples; all the rest are from P12 samples. Pre-secretory stage ameloblasts express *Col4a1*, *Col7a1*, and laminin α5 and β3 chains. Secretory and maturation stage ameloblasts express laminin α3, β3, and γ2 chains at a higher amount and express β1 and γ1 chains at a lower amount. Key: CL, cervical loop; od, odontoblasts.

**Figure 5 ijms-26-04134-f005:**
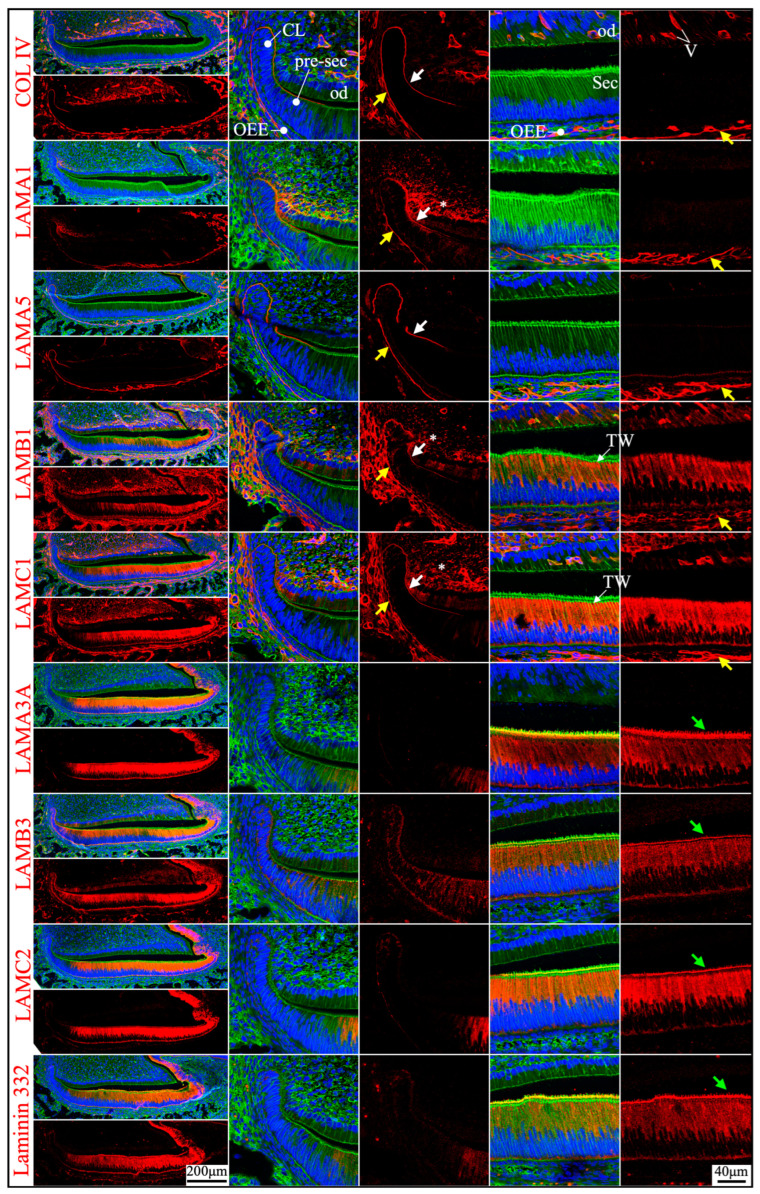
Immunohistochemistry of type IV collagen and laminin chains in 4-day-old mouse maxillary 1st molars. Type IV collagen (COL IV) and laminin chains are red. The cytoskeleton labeled by β-actin is green. Nuclei labeled by DAPI are blue. The basement membrane between the pre-secretory stage ameloblasts (Pre-sec) and odontoblasts (od) (white arrows) contains COL IV, LAMA1, LAMA5, LAMB1, and LAMC1. This basement membrane structure is degraded before the onset of the secretory stage. The basement membrane between the outer enamel epithelium (OEE) and dental follicle (yellow arrows) also contains COL IV, LAMA1, LAMA5, LAMB1, and LAMC1. The distal aspects of secretory stage ameloblasts (Sec) (green arrows) are labeled by LAMA3A, LAMB3, LAMC2, and laminin 332. The extracellular matrix of apical dental papilla cells (*) is positive for LAMA1, LAMB1, and LAMC1. LAMB1 and LAMC1 are detected in the supranuclear region of secretory stage ameloblasts, but the signal is not detected beyond the distal terminal webs (TW). Key: V, blood vessels; CL, cervical loop.

**Figure 6 ijms-26-04134-f006:**
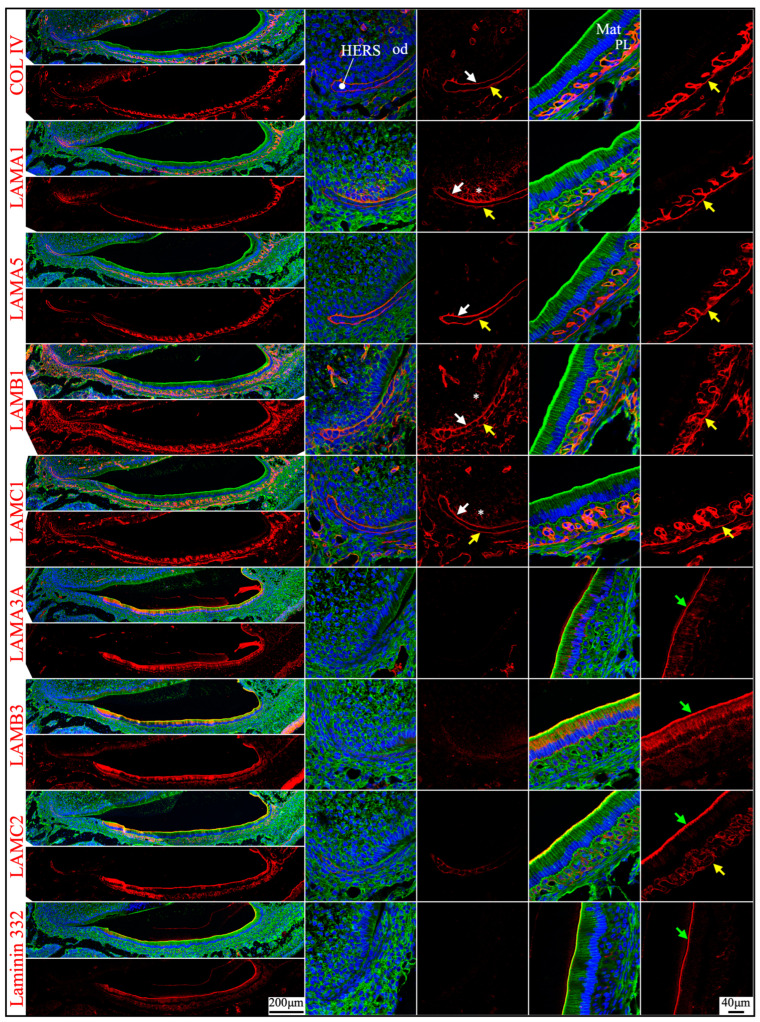
Immunohistochemistry of type IV collagen and laminin chains in 10-day-old mouse maxillary 1st molars. Type IV collagen (COL IV) and laminin chains are red. The cytoskeleton labeled by β-actin is green. Nuclei labeled by DAPI are blue. The basement membrane that encases the Hertwig’s epithelial root sheath (HERS) (white and yellow arrows) contains COL IV, LAMA1, LAMA5, LAMB1, and LAMC1. The basement membrane between HERS and odontoblasts (od) (white arrows) is degraded, while the basement membrane outside of HERS (yellow arrows) continues and becomes tortuous outside of the papillary layer (PL). This outer basement membrane (yellow arrows) also contains a low amount of LAMC2. The atypical basement membrane distal to the maturation stage ameloblasts (Mat) (green arrows) is positive for LAMA3A, LAMB3, LAMC2, and laminin 332. The extracellular matrix of apical dental papilla cells (*) is positive for LAMA1, LAMB1, and LAMC1.

Overall, the predominant laminin isoforms in the basement membrane structures at the interface both between the IEE and the dental papilla and between the OEE and the dental follicle were laminins 111 and 511.

### 2.2. Laminin 3A32 Were Localized Along the Secretory Face of Tomes’ Process During Secretory Stage Enamel Formation Without Forming a Basement Membrane Structure

As tooth formation progresses to the onset of dentin mineralization, the IEE differentiates into secretory stage ameloblasts that secrete a myriad of enamel matrix proteins that form the initial (aprismatic) enamel on the surface of dentin immediately after dentin mineral foci coalesce into a continuous mineral layer and while the secretory surface of the ameloblast distal membrane is in close proximity to the surface of dentin. The ameloblast distal membrane, while continuing to facilitate the enamel mineral ribbon elongation, changes its surface contour to generate a “Tomes’ process” with two incompletely separated secretory regions that facilitate “interrod” enamel ribbon extension over the intercellular junctions with the six surrounding ameloblasts and “rod” enamel from part of the protruding part of the Tomes’ process [15,43]. Although secretory ameloblasts do not exhibit a basement membrane structure at the mineralization front, they express high levels of *Lama3*, *Lamb3*, and *Lamc2*, a moderate level of *Lamc1* and a low level of *Lamb1* in both incisors and molars (Figure 1 and Figure 4). Type IV collagen, LAMA1, and LAMA5 signals were absent in the secretory stage ameloblasts (Figure 2, Figure 3 and Figure 5). LAMB1 (Figure 5) and LAMC1 (Figure 3 and Figure 5) signals were detected in the supranuclear region of secretory ameloblasts but were absent beyond the terminal webs. LAMA3A, LAMB3, LAMC2, and laminin 332 signals were strong in the cytoplasm (presumably along the secretory pathway) and along Tomes’ processes (Figure 3 and Figure 5). The non-secretory surface of Tomes’ processes was negative for laminin 332. In contrast, laminin 332 signals specifically localized along the secretory surface, corresponding to the putative rod and interrod growth sites [15] (Figure 3).

There are 76 exons in the mouse *Lama3* gene forming two transcript variants: *Lama3a* (NM_001347461.2) and *Lama3b* (NM_010680.2) (Figure 7A). The *Lama3a* transcript starts from exon 39 and goes through to exon 76. The *Lama3b* transcript starts from exon 1, skips exon 39, and continues through to exon 76. A forward primer targeting a region spanning exons 37 and 38, a forward primer targeting exon 39, and a reverse primer targeting exon 41 (Figure 7A) were designed to distinguish the *Lama3* transcripts in the secretory stage ameloblasts. mRNA was extracted from the enamel organ epithelium of 5-day-old mouse’s first molars, within which most ameloblasts are in the secretory stage. Reverse transcriptase polymerase chain reaction (RT-PCR) products using the mRNA as templates were subjected to electrophoresis. Amplicon was shown only when the forward primer targeting exon 39 was used (Figure 7B), indicating that only *Lama3a* but not *Lama3b* was expressed by secretory stage ameloblasts. This amplicon was cut out for Sanger sequencing using the reverse primer. Segments of the chromatogram demonstrated the reference sequence flanking the junction between exons 39 and 40 (Figure 7C). This was consistent with the LAMA3A immunolabeling, because this antibody specifically targeted the α3A chain N-terminal IIIa domain (amino acid positions 1–212 of LAMA3A) [40,41]. Therefore, the laminin isoform along the secretory surface of Tomes’ process was laminin 3A32.

**Figure 7 ijms-26-04134-f007:**
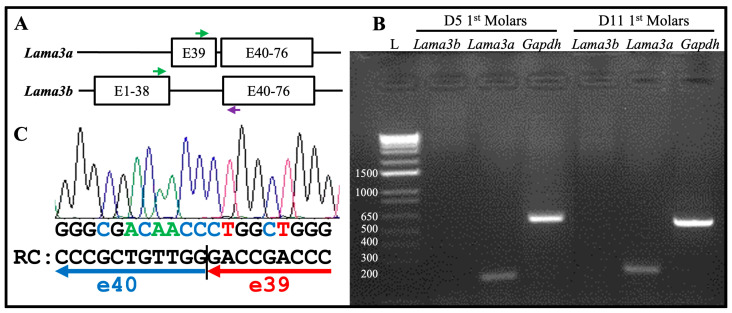
The *Lama3a* transcript, but not the *Lama3b* transcript, is expressed by the secretory and maturation stages of ameloblasts. (**A**) Schematic illustration of mouse *Lama3a* and *Lama3b* transcripts. The *Lama3a* transcript starts from exon 39 and goes through to exon 76. The *Lama3b* transcript starts from exon 1, skips exon 39, and continues through to exon 76. Green arrows: forward primers for RT-PCR in exon 39 (upper) and exons 37/38 (lower). Purple arrow: a common reverse primer for RT-PCR in exon 41. (**B**) Electrophoresis of the RT-PCR products using mRNA extracted from enamel organ epithelia of 5-day-old and 11-day-old mouse’s first molars that are mostly during secretory and maturation stages, respectively, as templates. L: DNA ladder annotated with the sizes of individual bands in bp. Only the forward primer targeting exon 39 and the common reverse primer produced amplicons of the expected size (193 bp). The *Gapdh* experimental control yielded a 530 bp amplicon. (**C**) The amplicon in (**B**) was excised for Sanger sequencing using the reverse primer. Part of the chromatography is shown. RC: the reverse complement sequence of partial chromatography matches the exon junction between exons 39 and 40.

### 2.3. The Atypical Basement Membrane of Maturation Stage Ameloblasts Contained Laminin 3A32

An electron-dense basement membrane is present along the distal surface of the maturation stage ameloblasts [37]. The maturation stage ameloblasts strongly express *Lama3*, *Lamb3*, and *Lamc2* (Figure 1 and Figure 4). LAMA3A, LAMB3, LAMC2, and laminin 332 were lucidly localized along the basement membrane of maturation stage ameloblasts (Figure 2, Figure 3 and Figure 6).

mRNA was extracted from the enamel organ epithelium of 11-day-old mouse’s first molars and RT-PCR experiments were conducted using the same sets of primers to distinguish the *Lama3* transcript in maturation stage ameloblasts. Amplicon was shown only when the forward primer targeting exon 39 was used (Figure 7B). This confirmed the laminin isoform along the maturation stage atypical basement membrane was laminin 3A32.

### 2.4. Junctional Epithelium Was Connected to the Enamel via Laminin 3A32 and to the Gingival Connective Tissues via Laminins 111 and 511

Junctional epithelium is derived from the reduced enamel organ and is continuous with oral sulcular epithelium and oral gingival epithelium [44]. The junctional epithelium has an external basal lamina (a basement membrane facing the connective tissue of gingiva) and an internal basal lamina (simple extracellular matrix facing the matured enamel) [44]. Laminin compositions of the two basal lamina structures were distinguished in the junctional epithelium around maxillary first molars in 21-day-old mice. LAMA1, LAMA5, LAMB1, and LAMC1 were primarily detected in the external basal lamina facing the gingival connective tissue (Figure 8), where LAMC2 and laminin 332 were detected intermittently. This distribution pattern was similar to the basement membrane on the outside of the papillary layer (Figure 3 and Figure 6), indicating that laminins 111 and 511 were the primary components of this basal lamina. In comparison, the internal basal lamina facing the enamel primarily contained LAMA3A, LAMB3, LAMC2, laminin 332, and possibly LAMC1 (Figure 8). This was similar to the atypical basement membrane on the distal side of maturation stage ameloblasts that was composed of laminin 3A32 (Figure 3 and Figure 6).

**Figure 8 ijms-26-04134-f008:**
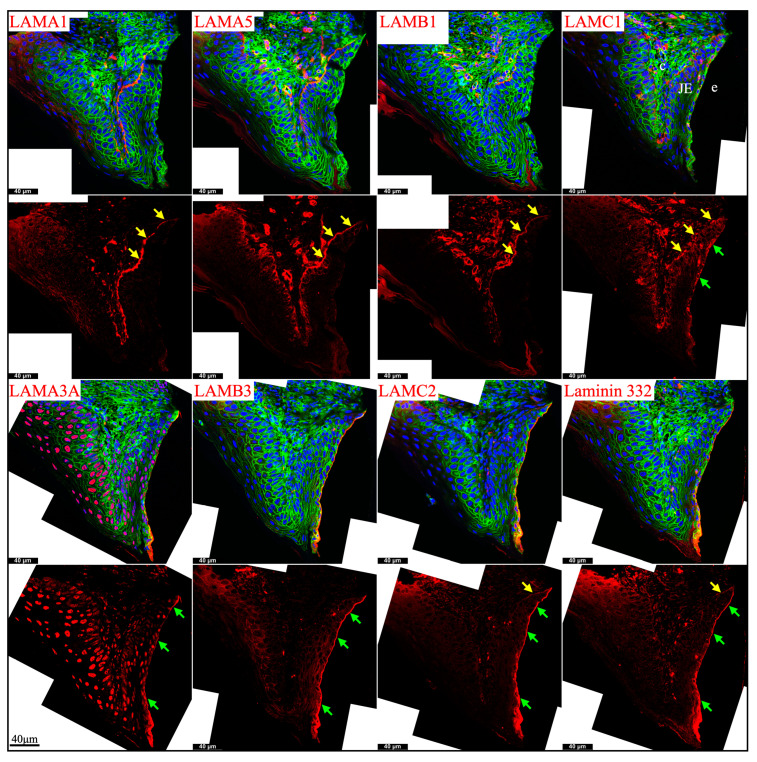
Immunohistochemistry of laminin chains in the junctional epithelium of 21-day-old mouse maxillary 1st molars. Laminin chains are red. The cytoskeleton labeled by β-actin is green. Nuclei labeled by DAPI are blue. The specialized basement membrane between the junctional epithelium (JE) and enamel (e) surface (green arrows) contains LAMA3A, LAMB3, LAMC2, laminin 332, and possibly LAMC1. The basement membrane between the junctional epithelium (JE) and the underlying connective tissues (c) (yellow arrows) contains LAMA1, LAMA5, LAMB1, and LAMC1; this basement membrane structure is intermittently positive for LAMC2 and laminin 332. Non-specific signals in LAMA3 immunostaining are seen in the nuclei.

These results supported that the junctional epithelium is derived from the reduced enamel organ epithelium.

### 2.5. Laminin 411 Was in the Endothelial Basement Membrane of Capillaries That Supplied the Dental Papilla and the Enamel Organ

Vascular supplies outside the enamel organ and inside the dental papilla ensure the nutrition and metabolic demands of the developing tooth organ [45]. Laminins 411 and 511 are predominant laminin isoforms in the endothelial basement membrane [46]. Blood vessels enter the dental papilla from the apical foramen in mouse incisors and from the openings surrounded by cervical loops in mouse molars and branch into capillaries that reach the odontoblast layer. The basement membrane surrounding these blood vessels and capillaries was positive for type IV collagen, LAMA4 (strongly), LAMA5 (weakly), LAMB1, and LAMC1 (Figure 9, white arrows). Thus, the dental papilla endothelial basement membrane primarily contained laminins 411 and 511, alongside type IV collagen.

**Figure 9 ijms-26-04134-f009:**
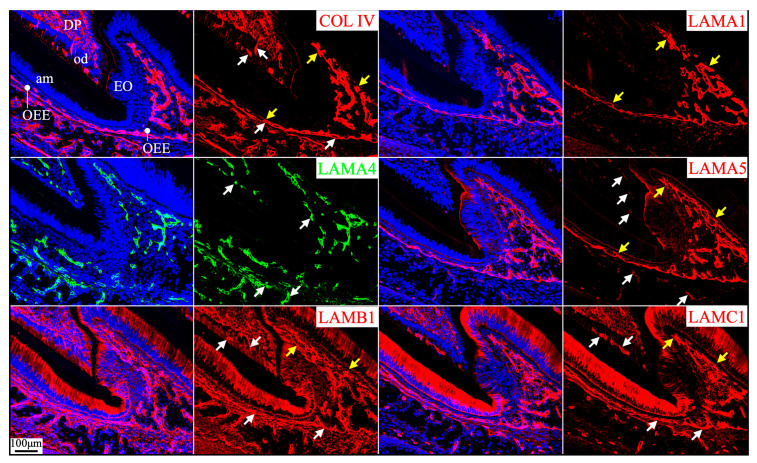
Immunohistochemistry of type IV collagen and laminin chains around the mesial cusp of 4-day-old mouse maxillary 1st molars. Collagen type IV (COL IV) and laminin chains are red or green as labeled. Nuclei labeled by DAPI are blue. The endothelial basement membrane of the capillaries (white arrows) supplying dental pulp (DP) and the enamel organ (EO) epithelium contains COL IV, LAMA4, LAMA5, LAMB1, and LAMC1. The basement membrane on the outer surface of the invaginated enamel organ epithelium that is continued from the cervical loop (yellow arrows) contains LAMA1, LAMA5, LAMB1, and LAMC1. Key: od, odontoblasts; am, ameloblasts; OEE, outer enamel epithelium.

The capillaries outside of the enamel organ indent into the enamel organ, as outer enamel epithelium invaginates when tooth mineralization starts [47]. The indented capillaries reach the stratum intermedium of the enamel organ without penetrating them [48]. The respective basement membrane structures from the enamel organ and from the endothelium remain intact under electron microscopy, despite their remodeling as the capillaries indent into the enamel organ [48]. The laminins 111 and 511 delineated the outer surface of enamel organ epithelium continuously from the apical loop to the incisal edge on longitudinal mouse mandibular incisor sections (Figure 2). However, the continuous boundary of the outer surface of the enamel organ from the cervical loops to the cusps in molars was only shown on the mesial and distal sides but not between cusps on a single sagittal molar section because of the complex contour of the enamel organ epithelium (Figure 9, yellow arrows). The concomitant enamel organ invagination and the capillary indentation made it difficult to distinguish the components of the two basement membrane structures. LAMA1 and LAMA5 were detected continuously and intensively from the cervical loop or HERS to the outer surface of the enamel organ (Figure 5 and Figure 6; Figure 9, yellow arrows), but LAMA4 was not (Appendix A). LAMA4 was detected at a comparable intensity between the capillaries surrounding the tooth organ and the indented capillaries (Figure 9, white arrows). In comparison, the LAMA5 signal potentially associated with capillaries was weaker (Figure 9, white arrows) than the LAMA5 signal in the OEE (Figure 9, yellow arrows). LAMB1 and LAMC1 were presumably in both basement membrane structures (Figure 9). These results suggested laminin 411 as a major component of the endothelial basement membrane of the indented capillaries, while laminin 511 as a minor component.

### 2.6. Laminin 111 Was Found in the Extracellular Matrix of Apical Dental Papilla Cells Without Forming a Basement Membrane Structure

Apical dental papilla cells are dental papilla cells slightly distal to the apical loop of mouse incisors or cervical loops of mouse molars that are relatively early in their differentiation status [49]. They correspond to pre-odontoblasts and adjacent dental pulp cells [50]. These cells may give rise to odontoblasts and the dental pulp lineage [49]. Apical dental papilla cells were actively proliferating, as they intensively immunolabeled for Ki67 (Figure 10). Ki67 expression abruptly decreased as differentiation progressed and was only sporadically detected in the distal dental papilla and completely absent in differentiated odontoblasts. *Col7a1*, *Lama1*, *Lamb1*, and *Lamc1* were moderately expressed by these proliferating dental papilla cells (Figure 1). In both incisors and molars, LAMA1, LAMB1, and LAMC1 signals were localized in the extracellular matrix of the apical dental papilla cells (Figure 2, Figure 3, Figure 5 and Figure 10). Although a basement membrane structure was absent around apical dental papilla cells, laminin 111 was a component of their extracellular matrix.

**Figure 10 ijms-26-04134-f010:**
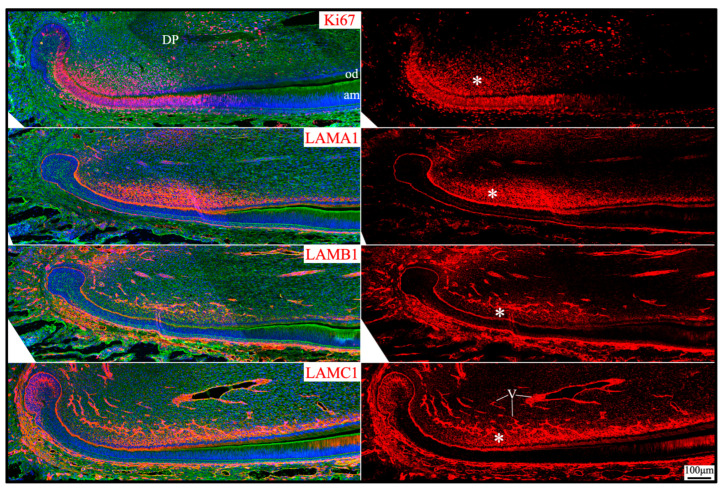
Immunohistochemistry of Ki67 and laminin chains in the apical dental papilla of 10-day-old mouse mandibular incisors. Ki67 signal (proliferating cells) and laminin chains are red. The cytoskeleton labeled by β-actin is green. Nuclei labeled by DAPI are blue. The extracellular matrix of apical dental papilla cells (*) that are actively proliferating is positive for LAMA1, LAMB1, and LAMC1. Key: DP, dental papilla; od, odontoblasts; am, ameloblasts; V, blood vessels.

## 3. Discussion

In the present study, we provide a comprehensive survey of expression and localization of individual laminin chains to establish the foundation for functional investigations of laminins in tooth development. RNAscope has become the gold standard for *in situ* hybridization because of its sensitivity and specificity [51]. We conducted RNAscope analyses of all 11 laminin chains, together with *Col4a1* and *Col7a1*, to reveal their expression patterns in developing mouse teeth. Additionally, we performed immunohistochemistry to localize laminin chains using antibodies, which provide location and intensity information that can be informative to their functions during tooth development. Antibody reliability can be undermined by cross-reactions, batch-to-batch variations, and unsuitable applications [52,53]. The best strategy to validate immunohistochemistry findings is to include a negative control from knockout mouse models for individual tissues [54], which is not always feasible. Alternatively, we (1) use antibodies that have been validated in other tissues [39,40], (2) rationalize signals from the known and plausible functions of the tested proteins, and (3) cross-validate findings from immunohistochemistry with those from *in situ* hybridization. All three strategies were used to make the best possible interpretations for signals observed in this study.

The basement membrane structures between the inner enamel epithelium (IEE) and dental papilla and between the outer enamel epithelium (OEE) and dental follicle are similar in their components. They both contain type IV collagen and laminins 111 and 511. Of note, the expression of *Lama5* is strong in IEE cells before they differentiate into secretory stage ameloblasts, as well as within OEE cells. However, the expression of *Lama1*, *Lamb1*, and *Lamc1* is relatively weak, particularly in the IEE cells. Nevertheless, signals of LAMA1, LAMA5, LAMB1, and LAMC1 along the distal membrane of differentiating IEE are clear. The discrepancy between the RNA and protein signals may be due to several factors: the sensitivity of the *Lama1* riboprobe and the differences in transcriptional and translational regulations of *Lama1*, *Lamb1*, *Lamc1* and *Lama5*. The basement membrane surrounding the enamel organ that contains LAMA5 appears to be essential for the proper proliferation and differentiation of the enamel organ, as suggested by the findings in *Lama5* null mice [17].

Regardless of their similar components, the turnover of basement membrane structures immediately adjacent to IEE and OEE is different. The basement membrane adjacent to IEE is degraded before the onset of enamel matrix deposition and mineral ribbon formation. The degradation of this presecretory basement membrane is closely associated with the onset of amelogenin expression [12]. Morphologically, numerous small protrusions that form at the distal membrane of ameloblasts are in direct contact with the predentin collagenous extracellular matrix [14]. Sometimes, termini of odontoblastic processes reach ameloblasts [14]. When initial dentin mineral foci appear in predentin matrix vesicles, they are immediately adjacent to the ameloblast distal membrane [13,14]. These findings suggest that the proper degradation of this presecretory basement membrane is essential for reciprocal interactions and subsequent dentin and enamel mineralization. The degradation of the presecretory basement membrane that contains laminins 111 and 511 is soon replaced by a basement membrane “invisible” under microscopes (no lamina densa) between the secretory stage ameloblast membrane and the elongating enamel ribbons that contains laminin 3A32. Basement membranes are constantly remodeled during development [55]. For example, the glomerular basement membrane transitions from laminin 111 to 511 and then 521 during its maturation [56]. In hair follicles, laminins 511 and 332 are differentially expressed and correlate distinctly with different phases of hair growth [57].

In contrast to the IEE presecretory basement membrane, the basement membrane adjacent to OEE is not degraded but seems to undergo constant remodeling of its contour. It is relatively linear before dentin and enamel mineralization starts on the other side of the enamel organ epithelium. The onset of mineralization blocks subsequent nutritional supplies from blood vessels in the dental papilla [47]. Thus, the invagination of the outer boundary of the enamel organ is accompanied by the indentation of blood vessels [48], which allows ample ions, nutrition and metabolites to be transported into the enamel organ to support enamel formation.

It remains unclear how laminin 3A32 participates in enamel formation. *LAMA3* and *LAMB3* biallelic human mutations display hypoplastic enamel [18,19,20,21], while *LAMA3* [22,23,24] and *LAMB3* [25,26,27,28,29] heterozygous mutations cause enamel pits and grooves in humans. Additionally, *Lama3* null mice show the detachment of ameloblasts [30]. Genetic defects in *Lamc2* also cause severe defects in the interface between maturation stage ameloblasts and maturing enamel [58]. Previously, it was perceived that laminin 332 was a component of the presecretory basement membrane [29,33]. However, LAMA3, LAMB3, LAMC2, and laminin 332 were detected in the secretory stage ameloblasts [31,34,35,36]. We proposed a molecular model that guides enamel ribbon formation at the enamel mineralization front based on findings from human genetics, knockout mouse phenotypes, and the expression and localization profiles of these molecules, which include laminin 332 [13]. In the present study, we further demonstrate that laminin 3A32, not 3B32, is localized along the secretory surface of Tomes’ processes of secretory stage ameloblasts. During the secretory stage of enamel formation, a lamina densa structure is not seen microscopically between the distal plasma membrane of ameloblasts and the elongating enamel ribbons, but a 25–30 nm clear space is maintained at this interface [59]. This space is similar to the typical thickness of a lamina lucida. Thus, we speculate that laminin 3A32 is localized within this electron-lucent area between the ameloblast and enamel ribbons to guide the appositional growth of the enamel ribbons. Of note, laminin 3A32 is special in its structure, featuring three truncated short arms. Laminin α3A, α4, and γ2 have much shorter N-terminal arms compared to the other laminin chains [60]. Particularly, laminin α3A, β3, and γ2 chains lack some or all of the N-terminal globular domains that are present in the other laminin chains [61]. Compared to the three short arms of laminins 111 and 3B32, the three short arms of laminin 3A32 appear to form a condensed globular structure [61]. The three short arms inhibit laminin 3A32 self-assembly [62]. Thus, laminin 3A32 along the distal membrane of secretory stage ameloblasts may not assemble to form a network. Additionally, laminin 3A32 may be anchored to the secretory surface of Tomes’ processes of ameloblasts via the binding between the C terminus of the laminin α3A chain and transmembrane integrin α6β4 [3,8]. Integrin α6β4 is a component of the hemidesmosome. Integrin α6β4 binds laminin 332 extracellularly and forms the hemidesmosome complex with type XVII collagen, plectin, and dystonin intracellularly to connect to the cytoskeleton [63]. Of note, developmental enamel defects were observed in human patients with genetic defects in the *ITGB4* gene that encodes the integrin β4 chain [64,65]. We speculate that each laminin 3A32 molecule interacts with enamel matrix proteins, such as enamelin and ameloblastin, to guide the elongation of an individual enamel ribbon during the secretory stage [13]. We further speculate that laminin 3A32 acts as a critical factor for the extracellular to intracellular communications of secretory stage ameloblasts through the hemidesmosome and cytoskeleton.

There is an atypical basement membrane that mediates a strong adhesion of the maturation stage ameloblasts to the maturing enamel surface. Amelotin (AMTN), odontogenic ameloblast-associated (ODAM), and laminin 332 have been identified in this basement membrane [37,66], while collagen types IV and VII are absent. We further showed that the laminin isoform in this atypical basement membrane is laminin 3A32. This atypical basement membrane structure is further present between the junctional epithelium and the matured enamel surface surrounding the cervical region of the tooth crown [67]. Our findings that laminin 3A32 is present along the internal basal lamina (tooth-facing junctional epithelium surface) and that laminins 111 and 511 are present along the external basal lamina (connective tissue-facing junctional epithelium surface) are consistent with the previous findings [38,68,69]. Together with hemidesmosomes and potentially secretory calcium-binding phosphoprotein proline-glutamine rich 1 (SCPPPQ1), laminin 3A32 enhances the attachment between the junctional epithelium and the hardened enamel surface, which is fundamental to periodontal health [70].

The interface between the endothelium and the enamel organ epithelium is an example of two juxtaposed basement membrane structures. They are derived from different developmental origins and are composed of distinctive laminin components. Extensive invaginations of the outer enamel epithelium border and indentations of the capillary system occur to supply the previously avascular enamel organ when dentin and enamel mineralization starts and the nutrition supplies from the dental papilla to ameloblasts become sparse [45,47]. Both the endothelium basement membrane and the outer enamel epithelium basement membrane remain intact during the invagination under electron microscopy [48,71]. In the present study, we show that laminins 111 and 511 in the outer enamel epithelium basement membrane are continuously present from the cervical loop to the outer surface of the enamel organ surrounding the cusps in mouse molars and from the apical loop to the outer surface of the junctional epithelium facing the connective tissue in mouse incisors. We also show that laminin 411 and potentially laminin 511 from the endothelium basement membrane follow the capillary indentations into the enamel organ. Similar juxtaposed double membrane structures can be found elsewhere. In the blood–brain barrier, the endothelial basement membrane containing laminins 411 and 511 and the glial basement membrane containing laminins 111 and 211 are in proximity to form a selective barrier [72].

Laminins are also detected in the extracellular matrix where a planar basement membrane structure is not observed, in addition to those along the secretory surface of Tomes’ processes. Laminins in the stem cell niche may interact with integrins on the cell surface to maintain their expansion and self-renewal activities [73,74]. Laminin 332, most likely laminin 3B32 (because it is negative for LAMA3A), is detected in the extracellular matrix of the apical loop but not in the basement membrane that encases the enamel organ epithelium. Laminin 332 may regulate the activities of dental epithelial stem cells. Laminins influence cell adhesion and migration, and cell type-specific laminins may drive cellular differentiation into various lineages [75]. The apical dental papilla cells are highly proliferative and may give rise to both dental pulp and odontoblast lineages [49]. Laminin 111 is detected in the extracellular spaces of the apical dental papilla without forming a visible basement membrane structure. We speculate that laminin 111 in the extracellular spaces may be critical to maintaining the bipotent differentiation capacity of the apical dental papilla cells. A similar scenario is found in articular cartilage chondrocytes, where laminins are condensed to a pericellular zone without forming a typical planar basement membrane structure [76]. Notably, individual laminin chains localize in different areas of this cartilage and are remodeled throughout development [76].

Other than laminin 111 in the extracellular matrix of apical dental papilla cells, other laminins are detected in the dental mesenchyme. *Lamb2* expression was low in dental epithelium but relatively high in dental mesenchyme (Appendix A). For the first time, the characteristic expression pattern of *Lamc3* was revealed in developing teeth and associated with the dental follicle and its derivative tissues (Appendix A). To date, the only known laminin isoforms that contain LAMC3 are laminins 423 and 523 [9]. *Lama4* and *Lamb2* transcripts were also detected in the dental follicle and its derivative tissues (Appendix A), suggesting that laminin 423 is in the dental follicle. Indeed, many laminin chains, including α2, α4, β1, β2, γ1, and γ3, are diffusely expressed by dental mesenchyme (both dental papilla and dental follicle) (Appendix A). The roles of these laminin chains in dental mesenchyme are unknown. It appears that LAMA2 is essential for odontoblast differentiation [77].

There are limitations in this study. While the specificity of antibodies was prioritized for the reliability of immunohistochemical signals, the sensitivity of antibodies may be insufficient for the detection of laminin chains with low expression levels or diffuse distribution patterns in the extracellular matrix. Furthermore, we were not able to obtain reliable antibodies for LAMB2 and LAMC3. Laminin 521 is a component of the epidermal basement membrane [78], but we are not able to determine whether LAMB2 is in the basement membrane surrounding the enamel organ epithelium.

In summary, we characterized the expression of all laminin chains and immunolocalized most of them in developing mouse teeth. The roles of laminins in developing teeth are likely multifaceted. Particularly, laminins provide a protective environment for the enamel organ during its proliferation and differentiation, mediate vascular structure invagination, facilitate epithelial and enamel surface adherence, and probably directly participate in the epithelium-controlled membrane-associated mechanism of enamel mineralization. Laminins are also present in the extracellular matrix without forming a planar basement membrane structure, which might be associated with the differentiation potentials of cells within that extracellular matrix. Overall, the localization of multiple laminin isoforms in developing teeth reflects the functional diversity that is established in other systems, including creating tissue separation and barriers, facilitating cell adhesion and migration, establishing cellular polarity, and regulating tissue morphogenesis and signaling [6].

This study provides a molecular mapping of laminins during tooth formation and enamel mineralization and establishes a foundation for functional investigations of laminins in organogenesis. The best way to reveal the *in vivo* functions of laminins in tooth development is to study the cellular, molecular, and ultrastructural changes in conditional knockout animal models of individual laminin chains. Currently, we are investigating the roles of laminin 3A32 in enamel mineralization using *Lama3* conditional knockout mouse models. It will also be interesting to study the functions of laminins 111 and 423 in the more heterogeneous dental mesenchyme.

## 4. Materials and Methods

### 4.1. Mice

All mice used in this study were housed in the Association for Assessment and Accreditation of Laboratory Animal Care International-accredited facilities. They were treated humanely in accordance with protocols approved by the University of Michigan Institutional Animal Care and Use Committees. The experimental procedures were carried out in compliance with ARRIVE guidelines.

### 4.2. Sample Preparation

Heads and hemimandibles were dissected from newborn, 4-day-old, 10-day-old, 12-day-old, 17-day-old, and 21-day-old wild-type C57BL/6 mice. Most ameloblasts in maxillary first molars of newborn, 4-day-old, and 10/12-day-old mice were in presecretory, secretory, and maturation stages, respectively. Mouse maxillary first molars erupted around postnatal day 15. The maxillary first molars of 17- and 21-day-old mice were used for the investigations of laminin isoforms adjacent to the junctional epithelium. The continuously growing mouse mandibular incisors erupted around postnatal day 10; thus, 10-day-old mandibular incisors that contained all developmental stages were used for the investigations of laminin expression in developing incisors. Mandibular incisors from 10-day-old mice, not older mice, were chosen because RNA preservation in these samples was the best due to the shortest duration of decalcification.

These samples were fixed in 4% paraformaldehyde at 4 °C overnight, decalcified in diethyl pyrocarbonate-treated 16.52% ethylenediaminetetraacetic acid at 4 °C, dehydrated in gradient ethanol, cleared in xylene and embedded into paraffin. Sagittal sections at 5 μm were collected.

### 4.3. RNAscope In Situ Hybridization

RNAscope *in situ* hybridization was performed using RNAscope^TM^ 2.5 HD assay–Red (Advanced Cell Diagnostics, Neward, CA, USA). Manufacturers’ protocols 322452 (FFPE sample preparation and pretreatment) and 322360 (RNAScope 2.5 HD Detection Reagent-RED) were followed. After deparaffinization, sections were treated with 3% H_2_O_2_. Target retrieval was performed for 15 min in a steamer, and sections were then incubated in Protease Plus for 30 min at 40 °C. Hybridization was performed for 2 h at 40 °C, followed by the incubations of AMP1, AMP2, AMP3, and AMP4 reagents for 30, 15, 30, and 15 min, respectively, at 40 °C. The incubations of AMP5 and AMP6 reagents were lengthened to 45 min and 20 min at ambient temperature, respectively, after a consultation with the manufacturer. RED detection reagents were incubated for 8 to 10 min at ambient temperature. The sections were then counterstained, dried, and mounted. The riboprobes used in this study, as well as the riboprobe as a negative control, were listed in Appendix A. RNAscope using the negative control probe was performed on a section under the same condition as other sections from the same block to ensure that no non-specific signals were present. Images were taken using a Nikon Eclipse TE300 microscope and photographed using a Nikon DXM1200 digital camera (Minato City, Tokyo, Japan). For each riboprobe, RNAscope was performed on sections from 2 to 3 mice of each developmental stage. RNAscope results for each cell type for a particular developmental stage were compared between incisors and molars. All RNAscope was performed by T.L.

### 4.4. Immunohistochemistry

Sections were baked and rehydrated in xylene and gradient ethanol. Antigen retrieval was performed in pH 9.0 Tris-EDTA buffer (10 mM Tris base, 1 mM Ethylenediaminetetraacetic acid, 0.05% Tween 20) in a steamer for 12 min, then cooled down at ambient temperature for 20 min. Sections were then blocked in 3% bovine serum albumin, 10% normal serum (goat or donkey, species depending on the host of the secondary fluorescent antibody), in phosphate-buffered saline containing 0.05% Tween 20 (PBST). Sections were then incubated with unconjugated primary antibodies, followed by the incubation in secondary fluorescent antibodies. All antibodies were diluted in 10% normal serum in PBST during incubation. In most cases, a β-actin antibody was used to label the cytoskeleton, followed by nuclear staining by 4′,6-diamidino-2-phenylindole (DAPI) and mounted in ProLong Diamond Antifade Mountant (Invitrogen, P36964; Waltham, MA, USA). All antibodies used in this study were listed in Appendix A. Images were taken using a Leica STELLARIS 8 confocal microscope with HyD detectors (Wetzlar, Hesse, Germany) at the Imaging Laboratory of the Michigan Diabetes Research Center. Most antibodies used in this study were validated by previous studies [39,40]. Immunohistochemical signals were cross-referenced with RNAscope signals for confirmation. Immunohistochemical signals were cross-validated between incisors and molars for cells in a corresponding developmental stage. All immunohistochemistry was performed by T.L. and M.S.

### 4.5. Reverse Transcriptase Polymerase Chain Reaction (RT-PCR)

RNA was extracted from the enamel organ epithelium of maxillary and mandibular first molars of 5-day-old and 11-day-old wild-type mice. Reverse transcription (RT) was conducted using the SuperScript III First-Strand Synthesis System (Invitrogen, 18080051; Waltham, MA, USA), and polymerase chain reaction (PCR) was performed using the Platinum Hot Start PCR Master Mix, 2X (Invitrogen, 13000012; Waltham, MA, USA). These primers were used: *Lama3*.e37.F: 5′-GATTGAGGGAAACTTCAGAC-3′; *Lama3*.e39.F: 5′-ATTTCTTCAGCGACCAAGTC-3′; *Lama3*.e41.R: 5′-GTTGTGCTGACAGTTAATGC-3′; *Gapdh* F: 5′-AGGCCGGTGCTGAGTATGTC-3′ and *Gapdh* R: 5′-TGCCTGCTTCACCACCTTCT-3′. Reaction conditions were denaturation @ 94 °C for 2 min, then [25 cycles of 94 °C for 30 s (template denaturation), then 58 °C for 30 s (primer annealing), followed by 72 °C for 50 s (primer extension)], 72 °C for 1 min and then hold at 4 °C. RT-PCR products were subjected to electrophoresis on a 2% agarose gel. *Lama3*.e37.F and *Lama3*.e41.R, *Lama3*.e39.F and *Lama3*.e41.R, and *Gapdh* F and *Gapdh* R were expected to produce amplicons of 367 bp, 193 bp, and 530 bp, respectively. Amplicons were then isolated for Sanger sequencing using the common *Lama3*.e41.R primer.

## 5. Conclusions

A comprehensive survey of the expression of all laminin chains and the localization of most of them in developing mouse teeth is shown in the current study. Primary laminin isoforms in basement membranes along the inner enamel epithelium before the secretory stage and outside of the outer enamel epithelium are laminins 111 and 511. Laminin 3A32 is present along the secretory surface of the secretory stage ameloblast Tomes’ processes, the atypical basement membrane of maturation stage ameloblasts, and the specialized basement membrane of junctional epithelium facing the enamel surface. The endothelial basement membrane in the dental papilla and outside of the enamel organ contains laminins 411 and 511. Laminin 332 is in the extracellular matrix, but not the basement membrane of the apical loop. Laminin 111 is in the extracellular matrix of the apical dental papilla without forming a visible basement membrane.

## Data Availability

Data are contained within the article and Appendix A.

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
