# Peer review of "Localizations of Laminin Chains Suggest Their Multifaceted Functions in Mouse Tooth Development"

_ijms, 2025, doi:10.3390/ijms26094134_

Round 1

Reviewer 1 Report

Comments and Suggestions for Authors

Dear authors, 

The topic covered in the article is very interesting and innovative, but the article is missing some important parts.

The introduction is lacking in content.

The materials and methods should be described before the results. There is also a lack of the discussion part where a comparison is made between the article and articles already present in scientific literature.

The limitations of the studies are missing and the conclusions are insufficient and poor in content, not fully answering the requirements of the article.

Without these important parts the article does not merit publication.

Author Response

We would like to thank the reviewers and editors for their thorough review and constructive suggestions for our manuscript. We find these comments very helpful for us to improve our manuscript, particularly the readability for our potential readers. We are sincerely appreciative of this. Please find our responses and revisions according to the reviewers’ suggestions in a point-by-point manner as followed.

Comment 1: The topic covered in the article is very interesting and innovative, but the article is missing some important parts.

The introduction is lacking in content.

Response 1: Thank you so much for pointing out the issue with our introduction session. I have added three points to the introduction session.

Basal lamina, sometimes used interchangeably with basement membrane, includes lamina lucida and lamina densa, but not lamina reticularis [5].

Previously, all 5 laminin α chains were found to be expressed during early molar morphogenesis [31]. LAMA1 and LAMA5 were localized along the basement membrane separating the dental epithelium and mesenchyme, while the two Lama3 variants ap-peared to be differentially expressed by cells in enamel organ epithelium [31,32]. There was controversy about whether Laminin 332 (a heterotrimer of laminin α3, β3, γ2 chains) was in the basement membrane separating dental epithelium and mesenchyme. Laminin 332 was localized along the presecretory basement membrane [33]. However, Lama3, Lamb3, and Lamc2 transcripts were detected in secretory ameloblasts [34-36], although a basement membrane structure was absent. Laminin 332 was localized along the atypical basement membrane along the distal surface of maturation ameloblasts [37] and the specialized basement membrane between the junctional epithelium and the matured enamel surface [38]. Particularly, the Lama3 variants that were expressed by ameloblasts during enamel mineralization were largely unclear. Considering the structural diversity of laminin chains (even between α3A and α3B chains) and the functional diversity of laminin isoforms, it is critical to comprehensively characterize the expression of all laminin genes and the localization of all laminin isoforms in developing teeth. This will lay the foundations for us to decipher the molecular mechanisms of dental, partic-ularly enamel, defects associated with laminins.

We hypothesize that laminin chains are differentially expressed in developing teeth, and that multiple laminin isoforms are differentially distributed in the extracellular matrix of developing teeth.

Hope these are sufficient.

Comment 2: The materials and methods should be described before the results. There is also a lack of the discussion part where a comparison is made between the article and articles already present in scientific literature.

Reponse 2: Thank you very much for your suggestions. We have moved the materials and methods session (now session 2) to before the results session (now session 3).

To clearly compare the previous findings in the literature to our findings, in addition to the discussion session, we added a paragraph in the introduction session to clearly delineate what has been in the literature.

Previously, all 5 laminin α chains were found to be expressed during early molar morphogenesis [31]. LAMA1 and LAMA5 were localized along the basement membrane separating the dental epithelium and mesenchyme, while the two Lama3 variants ap-peared to be differentially expressed by cells in enamel organ epithelium [31,32]. There was controversy about whether Laminin 332 (a heterotrimer of laminin α3, β3, γ2 chains) was in the basement membrane separating dental epithelium and mesenchyme. Laminin 332 was localized along the presecretory basement membrane [33]. However, Lama3, Lamb3, and Lamc2 transcripts were detected in secretory ameloblasts [34-36], although a basement membrane structure was absent. Laminin 332 was localized along the atypical basement membrane along the distal surface of maturation ameloblasts [37] and the specialized basement membrane between the junctional epithelium and the matured enamel surface [38]. Particularly, the Lama3 variants that were expressed by ameloblasts during enamel mineralization were largely unclear. Considering the structural diversity of laminin chains (even between α3A and α3B chains) and the functional diversity of laminin isoforms, it is critical to comprehensively characterize the expression of all laminin genes and the localization of all laminin isoforms in developing teeth. This will lay the foundations for us to decipher the molecular mechanisms of dental, partic-ularly enamel, defects associated with laminins.

Comment 3: The limitations of the studies are missing and the conclusions are insufficient and poor in content, not fully answering the requirements of the article.

Response 3: Thank you for pointing out the deficiency of these two sessions. Regarding the limitations of this study, we added the following to the discussion session.

There are limitations in this study. While the specificity of antibodies was prioritized for the reliability of immunohistochemical signals, the sensitivity of antibodies may be insufficient for the detections of laminin chains with low expression levels or diffuse distribution patterns in the extracellular matrix. Furthermore, we were not able to obtain reliable antibodies for LAMB2 and LAMC3. Laminin 521 is a component of the epidermal basement membrane [78], but we are not able to determine whether LAMB2 is in the basement membrane surrounding the enamel organ epithelium.

We also added a conclusion session to this manuscript.

A comprehensive survey of the expression of all laminin chains and the localization of most of them in developing mouse teeth is shown in the current study. Primary laminin isoforms in basement membranes along the inner enamel epithelium before secretory stage and outside of outer enamel epithelium are laminins 111 and 511. Laminin 3A32 is present along the secretory surface of secretory stage ameloblast Tomes’ processes, the atypical basement membrane of maturation stage ameloblasts, and the specialized basement membrane of junctional epithelium facing the enamel surface. Endothelial basement membrane in dental papilla and outside of enamel organ contains laminins 411 and 511. Laminin 332 is in the extracellular matrix, but not the basement membrane of apical loop. Laminin 111 is in the extracellular matrix of apical dental papilla without forming a visible basement membrane.

Hope these could resolve the concerns.

Comment 4: Without these important parts the article does not merit publication.

Response 4: By adjusting the manuscript arrangement and adding the above information to our manuscript, we hope the reviewer can reconsider their opinion of our manuscript. Again, we sincerely appreciate the reviewer’s efforts to help us improve our manuscript.

Reviewer 2 Report

Comments and Suggestions for Authors

Dear Authors,

The manuscript provides a thorough analysis of laminin expression during mouse tooth development. However, several areas require improvement:

1. Main Research Question and Relevance: The study effectively addresses the question of how different laminin isoforms are expressed and localized during mouse tooth development. This is an original and relevant topic, as it bridges the gap between extracellular matrix biology and dental morphogenesis. The findings are particularly valuable for understanding how basement membranes influence enamel formation and epithelial organization.

2. Methodology Refinement: The description of RNAscope in situ hybridization and immunohistochemistry procedures should be more detailed, particularly regarding probe validation and antibody specificity. Clarifications on negative controls and statistical analyses would strengthen the reliability of the results. It would also be beneficial to elaborate on the rationale behind choosing specific developmental stages and tissues for analysis.

3. Clarity and Readability: The text is highly technical, making it difficult to follow in some sections. Simplifying complex explanations and improving figure legends would enhance comprehension, particularly for a broader scientific audience. Additionally, the terminology used to describe basement membrane structures should be clearly defined and used consistently throughout the manuscript.

4. Data Presentation: While figures are well-constructed, some need better integration within the main text. A clearer connection between findings and their functional implications would strengthen the narrative. Some figures would benefit from additional annotations to highlight key structural elements, and the captions should provide more context on the significance of the displayed results.

5. Discussion Improvements: The discussion is extensive but lacks a clear distinction between novel findings and previously established knowledge. A more structured synthesis of key conclusions and their broader implications for developmental biology would be beneficial. Additionally, the functional significance of laminin 3A32 in ameloblasts should be more explicitly addressed, as this finding is a key contribution of the study.

6. Consistency of Conclusions: The conclusions align with the evidence and arguments presented, but the manuscript would benefit from a stronger synthesis of findings in relation to the broader field. Highlighting potential future research directions, such as functional studies on laminin-deficient models, would improve the impact of the study.

7. References: The references are mostly appropriate, but adding citations to prior studies that have explored the role of laminins in basement membrane biology and enamel formation would provide a more comprehensive context.

8. Figures and Tables: The figures effectively illustrate the experimental results, but some could be optimized for clarity. For instance, it would be helpful to ensure uniform labeling conventions and to provide more explicit explanations of staining patterns in immunohistochemistry images.

Addressing these issues will significantly improve the manuscript’s clarity, rigor, and impact.

Author Response

We would like to thank the reviewers and editors for their thorough review and constructive suggestions for our manuscript. We find these comments very helpful for us to improve our manuscript, particularly the readability for our potential readers. We are sincerely appreciative of this. Please find our responses and revisions according to the reviewers’ suggestions in a point-by-point manner as followed.

The manuscript provides a thorough analysis of laminin expression during mouse tooth development. However, several areas require improvement:

Comment 1. Main Research Question and Relevance: The study effectively addresses the question of how different laminin isoforms are expressed and localized during mouse tooth development. This is an original and relevant topic, as it bridges the gap between extracellular matrix biology and dental morphogenesis. The findings are particularly valuable for understanding how basement membranes influence enamel formation and epithelial organization.

Response 1: We appreciate that the reviewers recognized the value of this article.

Comment 2. Methodology Refinement: The description of RNAscope in situ hybridization and immunohistochemistry procedures should be more detailed, particularly regarding probe validation and antibody specificity. Clarifications on negative controls and statistical analyses would strengthen the reliability of the results. It would also be beneficial to elaborate on the rationale behind choosing specific developmental stages and tissues for analysis.

Response 2: Thank you very much for your suggestions! We have added details to the RNAscope and immunohistochemistry procedures. You will be able to find the highlighted changes in the manuscript. Specifically, regarding probe validation: “RNAscope using the negative control probe was performed on a section under the same condition as other sections from the same block to ensure that no non-specific signals were present”. Regarding antibody specificity: “Most antibodies used in this study were validated by previous studies [39,40]. Immunohistochemical signals were cross referenced with RNAscope signals for confirmation. Immunohistochemical signals were cross validated between incisors and molars for cells in a corresponding developmental stage”. Negative control for RNAscope was performed using a riboprobe targeted a bacterial sequence (DapB, neg ctrl) that does not target molecules in the wild-type mammalian tissues (information in Table S1). Since the purpose of this study is to investigate the expression and localization of laminin chains in developing teeth, there is no statistic tests performed; but we did repeat RNAscope and immunohistochesmitry in 2-3 mice of each developmental stage for each riboprobe and antibody to make sure all signals are reproducible. We also take additional measures as stated above, such as cross reference signals from immunohistochemistry and RNAscope to make sure the signals are reliable.

We also elaborate the rationale for choosing specific developmental stages: “Heads and hemimandibles were dissected from newborn, 4-day-old, 10-day-old, 12-day-old, 17-day-old, and 21-day-old wild-type C57BL/6 mice. Most ameloblasts in maxillary first molars of newborn, 4-day-old, and 10/12-day-old mice were in presecretory, secretory, and maturation stages, respectively. Mouse maxillary first molars erupted around postnatal day 15. The maxillary first molars of 17- and 21-day-old mice were used for the investigations of laminin isoforms adjacent to the junctional epithelium. The continuously growing mouse mandibular incisors erupted around postnatal day 10; thus, 10-day-old mandibular incisors that contained all developmental stages were used for our investigations of laminin expression in developing incisors. Mandibular incisors from 10-day-old mice, not older mice, were chosen, because RNA preservation in these samples were the best due to the shortest duration of decalcification.”

Comment 3. Clarity and Readability: The text is highly technical, making it difficult to follow in some sections. Simplifying complex explanations and improving figure legends would enhance comprehension, particularly for a broader scientific audience. Additionally, the terminology used to describe basement membrane structures should be clearly defined and used consistently throughout the manuscript.

Response 3: Thank you for your suggestions! We have added more annotations for figures 3, 5, 6, 8, 9, and revised the figure legends for all figures, hoping to make it easier for the readers. To precisely describe the localization of laminin chains, we also provided more information about the anatomical/histological structures in figure annotations and legends.

For terminology, we added a sentence to describe the differences between basement membrane and basal lamina in the introduction : “Basal lamina, sometimes used interchangeably with basement membrane, includes lamina lucida and lamina densa, but not lamina reticularis [5]”.We used basement membrane most of the time, except for the external and internal basal lamina on the two sides of junctional epithelium (session 3.4). We did this because 1) the internal basal lamina that is connected to the dental enamel may not contain a lamina reticularis; 2) we want to be consistent with the literature.

Comment 4. Data Presentation: While figures are well-constructed, some need better integration within the main text. A clearer connection between findings and their functional implications would strengthen the narrative. Some figures would benefit from additional annotations to highlight key structural elements, and the captions should provide more context on the significance of the displayed results.

Response 4: Thank you very much for your suggestions! We have added a paragraph at the beginning of the results to provide an overview of the presentations of our data:

“Here, our findings were presented with three major focuses: laminins associated with the enamel organ epithelium and its derivative (3.1-3.4 and Figures 1-8), the basement membrane of vascular supplies to tooth germs (3.5 and Figure 9), and laminin components in the extracellular matrix of apical dental papilla cells (3.6 and Figure 10). Regarding laminins associated with the enamel organ epithelium and its derivative, findings on basement membrane structures that are not directly associated with mineralization (3.1), laminins associated with secretory (3.2) and maturation (3.3) stages ameloblasts, and laminins in junctional epithelium (3.4) will be presented. Seminal findings of RNAscope and immunohistochemistry from incisors were present in Figures 1-3, while findings of RNAscope and immunohistochemistry from molars of different stages were present in Figures 4-6. RT-PCR results to distinguish the Lama3 transcript were shown in Figure 7. Immunohistochemical results on the junctional epithelium were shown in Figure 8.”

We have added more annotations for figures 3, 5, 6, 8, 9, and revised the figure legends for all figures, hoping to make it easier for the readers. To precisely describe the localization of laminin chains, we also provided more information about the anatomical/histological structures.

Comment 5. Discussion Improvements: The discussion is extensive but lacks a clear distinction between novel findings and previously established knowledge. A more structured synthesis of key conclusions and their broader implications for developmental biology would be beneficial. Additionally, the functional significance of laminin 3A32 in ameloblasts should be more explicitly addressed, as this finding is a key contribution of the study.

Response 5: Thanks for raising these concerns!

To make a clear distinction between information from literature and our new findings, we added one new paragraph in the introduction session that detail the previous findings.

“Previously, all 5 laminin α chains were found to be expressed during early molar morphogenesis [31]. LAMA1 and LAMA5 were localized along the basement membrane separating the dental epithelium and mesenchyme, while the two Lama3 variants ap-peared to be differentially expressed by cells in enamel organ epithelium [31,32]. There was controversy about whether Laminin 332 (a heterotrimer of laminin α3, β3, γ2 chains) was in the basement membrane separating dental epithelium and mesenchyme. Laminin 332 was localized along the presecretory basement membrane [33]. However, Lama3, Lamb3, and Lamc2 transcripts were detected in secretory ameloblasts [34-36], although a basement membrane structure was absent. Laminin 332 was localized along the atypical basement membrane along the distal surface of maturation ameloblasts [37] and the specialized basement membrane between the junctional epithelium and the matured enamel surface [38]. Particularly, the Lama3 variants that were expressed by ameloblasts during enamel mineralization were largely unclear. Considering the structural diversity of laminin chains (even between α3A and α3B chains) and the functional diversity of laminin isoforms, it is critical to comprehensively characterize the expression of all laminin genes and the localization of all laminin isoforms in developing teeth. This will lay the foundations for us to decipher the molecular mechanisms of dental, partic-ularly enamel, defects associated with laminins.”

We added a conclusion session to provide a more structured synthesis of the key findings and their implications. At the end of our discussion, we also added a paragraph to relate our findings to a broader audience in the field of developmental biology:

“In summary, we characterized the expression of all laminin chains and immunolocalize most of them in developing mouse teeth. The roles of laminins in developing teeth are likely multifaceted. Particularly, laminins provide a protective environment for the enamel organ during its proliferation and differentiation, mediate vascular structure invagination, facilitate epithelial and enamel surface adherence, and probably directly participate in the epithelium-controlled membrane-associated mechanism of enamel mineralization. Laminins are also present in the extracellular matrix without forming a planar basement membrane structure, which might be associated with the differentiation potentials of cells within that extracellular matrix. Overall, the localization of multiple laminin isoforms in developing teeth reflects the functional diversity that is established in other systems, including creating tissue separation and barrier, facilitating cell adhesion and migration, establishing cellular polarity, and regulating tissue morphogenesis and signaling [6].”

We also expanded the discussion on the functional significance of laminin 3A32 in ameloblasts:

“In the present study, we further demonstrate that laminin 3A32, not 3B32, is localized along the secretory surface of Tomes’ processes of secretory stage ameloblasts. … Thus, laminin 3A32 along the distal membrane of secretory stage ameloblasts may not assemble to form a network. Additionally, laminin 3A32 can be anchored to the secretory surface of Tomes’ processes of ameloblasts through the binding between the C terminus of laminin α3A chain and transmembrane integrin α6β4 [3,8]. Integrin α6β4 is a component of the hemidesmosome. Integrin α6β4 binds laminin 332 extracellularly, and form the hemidesmosome complex with type XVII collagen, plectin, and dystonin intracellularly to connect to the cytoskeleton [63]. Of note, developmental enamel defects were observed in human patients with genetic defects in the ITGB4 gene that encode the integrin β4 chain [64,65]. We speculate that each laminin 3A32 molecule interacts with enamel matrix proteins, such as enamelin and ameloblastin, to guide the elongation of an individual enamel ribbon during the secretory stage [13]. We further speculate that laminin 3A32 acts as a critical factor for the extracellular to intracellular communications of secretory stage ameloblasts through the hemidesmosome and cytoskeleton”.

We really appreciate your suggestions for the improvement of our manuscript.

Comment 6. Consistency of Conclusions: The conclusions align with the evidence and arguments presented, but the manuscript would benefit from a stronger synthesis of findings in relation to the broader field. Highlighting potential future research directions, such as functional studies on laminin-deficient models, would improve the impact of the study.

Response 6: Thank you! We added a conclusion session to provide a synthesis of our findings. We also highlight potential future research directions as followed:

“The best way to reveal in vivo functions of laminins in tooth development is to study the cellular, molecular, and ultrastructural changes of conditional knockout animal models of individual laminin chains. Currently, we are investigating the roles of laminin 3A32 in enamel mineralization using Lama3 conditional knockout mouse models. It will also be interesting to study functions of laminins 111 and 423 in the more heterogeneous dental mesenchyme”.

Comment 7. References: The references are mostly appropriate, but adding citations to prior studies that have explored the role of laminins in basement membrane biology and enamel formation would provide a more comprehensive context.

Response 7: Thank you for this suggestion! As previously stated in our answers to your point 5, we provided a comprehensive review of previous findings on laminins in enamel formation in the introduction session. We cited most of the previous papers in this paragraph. We hope this to be helpful.

Comment 8. Figures and Tables: The figures effectively illustrate the experimental results, but some could be optimized for clarity. For instance, it would be helpful to ensure uniform labeling conventions and to provide more explicit explanations of staining patterns in immunohistochemistry images.

Response 8: Thank you for raising this point! We have made consistent labeling and added more annotations to the anatomical/histological structures to our immunohistochemical images. We also provided more explicit explanations in our figure legend. This should be helpful to improve the clarity of our data.

Comment 9: Addressing these issues will significantly improve the manuscript’s clarity, rigor, and impact.

Response 9: We would like to sincerely thank you again for these constructive suggestions! Hopefully, the changes that we made have improved the clarity, rigor, and impact of our manuscript.

Reviewer 3 Report

Comments and Suggestions for Authors

Dear authors, thank you for submitting the manuscript "Localizations of lamini chains suggest their multifaceted functions in mouse tooth development". I carefully read it and here is my feedback:

-The iThenthicate report is 23%, please decrease it. The articles that have high similarity to yours are: https://doi.org/10.3390/ijms26041646, https://doi.org/10.3390/ijms26010157 and https://doi.org/10.3390/ijms26020622.

-Please decrease the self-citation of some authors, example Simmer has 8.

-Introduction can be improve, please add some more information.

-In the discussion section, please specify why your samples were dissected at 4-day-old, 10 days old, 12 days old, 17 days old and 21 days old. 

-In the Materials and methods, specify if the experimental procedures were performed by a single experienced researcher or the 9 the authors performed all procedures.

-Expand your speculation regarding how unclear lamina 3A32 participates in enamel formation.

-Mention your speculations regarding the roles of the lamini chains diffusely expressed by dental mesenchyme (dental papilla and follicle).

-State your limitations of the study at the end of the Discussion section.

-Mention what future studies you would like to perform based on your results.

-Please use passive voice when writing the article. For example, instead of saying 'we used all three strategies...' in line 403, you should say 'the three strategies were used...'

-Too many references for a study, this is not a systematic review, please decrease it.

-Please state your hypotheses at the end of the introduction section.

-For the results section, please insert a chart or graphic abstract with the steps of your study so readers can easily follow the sequence.

-According to the journal's template, your are missing section 5 which is Conclusion, please include it.

-Please make sure all references follow the required style.

-Also, I see some very old references from 1960, 1967, 1962, etc, and if it is possible please use novel/updated studies as references.

Author Response

We would like to thank the reviewers and editors for their thorough review and constructive suggestions for our manuscript. We find these comments very helpful for us to improve our manuscript, particularly the readability for our potential readers. We are sincerely appreciative of this. Please find our responses and revisions according to the reviewers’ suggestions in a point-by-point manner as followed.

Comment 1: Dear authors, thank you for submitting the manuscript "Localizations of lamini chains suggest their multifaceted functions in mouse tooth development". I carefully read it and here is my feedback:

Response 1: Thank you very much for reading our manuscript carefully!

Comment 2: The iThenthicate report is 23%, please decrease it. The articles that have high similarity to yours are: https://doi.org/10.3390/ijms26041646, https://doi.org/10.3390/ijms26010157 and https://doi.org/10.3390/ijms26020622.

Response 2: Thank you for checking iTenthicate report for us. Honestly, I don’t know how to improve from the iThenthicate report. The titles of these three articles are: 1) A Weapon Against Implant-Associated Infections: Antibacterial and Antibiofilm Potential of Biomaterials with Titanium Nitride and Titanium Nitride-Silver Nanoparticle Electrophoretic Deposition Coatings, 2) Recent Advances in the Search for Effective Anti-Alzheimer’s Drugs, and 3) Viability and Radiosensitivity of Human Tumor Cells from Breast and Colon Are Influenced by Hypericum perforatum Extract HP01. These three papers have nothing to do with our manuscript or topic. The only thing in common is that all of them were published in the Int. J. Mol. Sci. However, we have revised throughout the manuscript based on other suggestions. Hope this could help.

Comment 3: Please decrease the self-citation of some authors, example Simmer has 8.

Response 3: Thanks. We have decreased our self-citation paper. For example, now there are only 6 papers from Dr. Simmer. We have also included other citations parallelly with our self-citation to increase the credibility. For example: “Biallelic mutations in LAMA3, LAMB3, and LAMC2 cause junctional epidermolysis bullosa with severe enamel hypoplasia in humans [18-21], while single allelic defects in LAMA3 [22-24] and LAMB3 [25-29] cause localized enamel defects without skin fragility.”

Comment 4: Introduction can be improve, please add some more information.

Response 4: Thank you for your suggestion! We have added another paragraph to our introduction to thoroughly summarize the investigations of expression and functions of laminin chains in tooth development. We also added a hypothesis to our introduction.

“Previously, all 5 laminin α chains were found to be expressed during early molar morphogenesis [31]. LAMA1 and LAMA5 were localized along the basement membrane separating the dental epithelium and mesenchyme, while the two Lama3 variants appeared to be differentially expressed by cells in enamel organ epithelium [31,32]. There was controversy about whether laminin 332 (a heterotrimer of laminin α3, β3, and γ2 chains) was in the basement membrane separating dental epithelium and mesenchyme. Laminin 332 was localized along the presecretory basement membrane [33]. However, Lama3, Lamb3, and Lamc2 transcripts were detected in secretory ameloblasts [34-36], although a basement membrane structure was absent. Laminin 332 was localized along the atypical basement membrane along the distal surface of maturation ameloblasts [37] and the specialized basement membrane between the junctional epithelium and the matured enamel surface [38]. Particularly, the Lama3 variants that were expressed by ameloblasts during enamel mineralization were largely unclear. Considering the structural diversity of laminin chains (even between α3A and α3B chains) and the functional diversity of laminin isoforms, it is critical to comprehensively characterize the expression of all laminin genes and the localization of all laminin isoforms in developing teeth. This will lay the foundations for us to decipher the molecular mechanisms of dental, particularly enamel, defects associated with laminins.”

“We hypothesize that laminin chains are differentially expressed in developing teeth, and that multiple laminin isoforms are differentially distributed in the extracellular matrix of developing teeth.”

Comment 5: In the discussion section, please specify why your samples were dissected at 4-day-old, 10 days old, 12 days old, 17 days old and 21 days old. 

Response 5: Thank you for your suggestion! We included the rationale of choosing these ages in our Methods session. We did this because another reviewer suggested us to move the Methods session to the front. The information is as followed.

“Heads and hemimandibles were dissected from newborn, 4-day-old, 10-day-old, 12-day-old, 17-day-old, and 21-day-old wild-type C57BL/6 mice. Most ameloblasts in maxillary first molars of newborn, 4-day-old, and 10/12-day-old mice were in presecretory, secretory, and maturation stages, respectively. Mouse maxillary first molars erupted around postnatal day 15. The maxillary first molars of 17- and 21-day-old mice were used for the investigations of laminin isoforms adjacent to the junctional epithelium. The continuously growing mouse mandibular incisors erupted around postnatal day 10; thus, 10-day-old mandibular incisors that contained all developmental stages were used for our investigations of laminin expression in developing incisors. Mandibular incisors from 10-day-old mice, not older mice, were chosen, because RNA preservation in these samples were the best due to the shortest duration of decalcification.”

Comment 6: In the Materials and methods, specify if the experimental procedures were performed by a single experienced researcher or the 9 the authors performed all procedures.

Response 6: Thanks for your suggestion. Information was provided in the Author Contributions session. We have added the information to the Methods as well.

All RNAscope was performed by T.L.

All immunohistochemistry were performed by T.L. and M.S.

Comment 7: Expand your speculation regarding how unclear lamina 3A32 participates in enamel formation.

Response 7: Thanks for the opportunity to expand this discussion. We added more information to this topic.

“In the present study, we further demonstrate that laminin 3A32, not 3B32, is localized along the secretory surface of Tomes’ processes of secretory stage ameloblasts. … Thus, laminin 3A32 along the distal membrane of secretory stage ameloblasts may not assemble to form a network. Additionally, laminin 3A32 can be anchored to the secretory surface of Tomes’ processes of ameloblasts through the binding between the C terminus of laminin α3A chain and transmembrane integrin α6β4 [3,8]. Integrin α6β4 is a component of the hemidesmosome. Integrin α6β4 binds laminin 332 extracellularly, and form the hemidesmosome complex with type XVII collagen, plectin, and dystonin intracellularly to connect to the cytoskeleton [63]. Of note, developmental enamel defects were observed in human patients with genetic defects in the ITGB4 gene that encode the integrin β4 chain [64,65]. We speculate that each laminin 3A32 molecule interacts with enamel matrix proteins, such as enamelin and ameloblastin, to guide the elongation of an individual enamel ribbon during the secretory stage [13]. We further speculate that laminin 3A32 acts as a critical factor for the extracellular to intracellular communications of secretory stage ameloblasts through the hemidesmosome and cytoskeleton.”

Comment 8: Mention your speculations regarding the roles of the lamini chains diffusely expressed by dental mesenchyme (dental papilla and follicle).

Response 8: Thank you for your suggestion. We added the following information to our revised discussion.

“Other than laminin 111 in the extracellular matrix of apical dental papilla cells, other laminins are detected in the dental mesenchyme.Lamb2 expression was low in dental epithelium but relatively high in dental mesenchyme (Figure S9). For the first time, the characteristic expression pattern of Lamc3 was revealed in developing teeth and associated with dental follicle and its derivative tissues (Figure S13). To date, the only known laminin isoforms that contain LAMC3 are laminins 423 and 523 [9]. Lama4 and Lamb2transcripts were also detected in dental follicle and its derivative tissues (Figures S6&S9), suggesting that laminin 423 is in the dental follicle. Indeed, many laminin chains, including α2, α4, β1, β2, γ1, and γ3, are diffusely expressed by dental mesenchyme (both dental papilla and dental follicle) (Figures S3-S13). The roles of these laminin chains in dental mesenchyme are unknown. It appears that LAMA2 is essential for odontoblast differentiation [77].”

Comment 9: State your limitations of the study at the end of the Discussion section.

Response 9: Thanks for raising this point! We added the following limitations to our study.

“There are limitations in this study. While the specificity of antibodies was prioritized for the reliability of most immunohistochemical signals, the sensitivity of antibodies may be insufficient to detect laminins in the cases of low expression levels and diffuse distribution in the extracellular matrix. Furthermore, we were not able to obtain reliable antibodies for LAMB2 and LAMC3. Laminin 521 is a component of the epidermal basement membrane [78], but we are not able to determine whether LAMB2 is in the basement membrane surrounding the enamel organ epithelium.”

Comment 10: Mention what future studies you would like to perform based on your results.

Response 10: Thank you for your suggestion! We added this to the end of our discussion.

The best way to reveal in vivo functions of laminins in tooth development is to study the cellular, molecular, and ultrastructural changes of conditional knockout animal models of individual laminin chains. Currently, we are investigating the roles of laminin 3A32 in enamel mineralization using Lama3 conditional knockout mouse models. It will also be interesting to study functions of laminins 111 and 423 in the more heterogeneous dental mesenchyme.

Comment 11: Please use passive voice when writing the article. For example, instead of saying 'we used all three strategies...' in line 403, you should say 'the three strategies were used...'

Response 11: We have gone through our manuscript and revised most of these. Some of them are still not in passive voice because those are our speculations and opinions. Hope this is fine.

Comment 12: Too many references for a study, this is not a systematic review, please decrease it.

Response 12: Thank you, but sorry that we are eventually not able to do so. We did take out some reference from our original manuscript, for example, to reduce self-citation. But since all three reviewers suggested adding more information to the manuscript, many related to the literature, we are not able to achieve that without adding reference. Hope this is fine.

Comment 13: Please state your hypotheses at the end of the introduction section.

Response 13: Thank you for bringing up this important point. We have added our hypothesis as followed:

We hypothesize that laminin chains are differentially expressed in developing teeth, and that multiple laminin isoforms are differentially distributed in the extracellular matrix of developing teeth.

Comment 14: For the results section, please insert a chart or graphic abstract with the steps of your study so readers can easily follow the sequence.

Response 14: Thank you very much for this suggestion! We added a paragraph at the beginning of our results session to guide our readers through the manuscript.

Here, our findings were presented with three major focuses: laminins associated with the enamel organ epithelium and its derivative (3.1-3.4 and Figures 1-8), the basement membrane of vascular supplies to tooth germs (3.5 and Figure 9), and laminin components in the extracellular matrix of apical dental papilla cells (3.6 and Figure 10). Regarding laminins associated with the enamel organ epithelium and its derivative, findings on basement membrane structures that are not directly associated with mineralization (3.1), laminins associated with secretory (3.2) and maturation (3.3) stages ameloblasts, and laminins in junctional epithelium (3.4) will be presented. Seminal findings of RNAscope and immunohistochemistry from incisors were present in Figures 1-3, while findings of RNAscope and immunohistochemistry from molars of different stages were present in Figures 4-6. RT-PCR results to distinguish the Lama3 transcript were shown in Figure 7. Immunohistochemical results on the junctional epithelium were shown in Figure 8.

Comment 15: According to the journal's template, your are missing section 5 which is Conclusion, please include it.

Response 15: Thanks! We added a conclusion session to this manuscript.

  1. Conclusion

A comprehensive survey of the expression of all laminin chains and the localization of most of them are in developing mouse teeth is shown by the current study. Primary laminin isoforms in basement membranes along the inner enamel epithelium before secretory stage and outside of outer enamel epithelium are laminins 111 and 511. Laminin 3A32 is present along the secretory surface of secretory stage ameloblast Tomes’ processes, the atypical basement membrane of maturation stage ameloblasts, and the specialized basement membrane of junctional epithelium facing the enamel surface. Endothelial basement membrane in dental papilla and outside of enamel organ contains laminins 411 and 511. Laminin 332 is in the extracellular matrix, but not the basement membrane of apical loop. Laminin 111 is in the extracellular matrix of apical dental papilla without forming a visible basement membrane.

Comment 16: Please make sure all references follow the required style.

Response 16: Thanks! We removed the reference that could cause the style problem.

Comment 17: Also, I see some very old references from 1960, 1967, 1962, etc, and if it is possible please use novel/updated studies as references.

Response 17: Thanks for looking into this. When we cited paper, we hope to the cite the most recent and the best ones that provided the best evidence on a certain finding. For example, the topic of vascular supply to tooth germs was mostly investigated a long time ago. Only few papers on this topic can be found after 2000, and those were all with different focuses but did not provide sufficient evidence on a very specific finding. The Bernick, 1960 paper is the best one to give a wholistic view of vascular supply in teeth. The Decker, 1963 and 1967 papers provided the clearest and the only electron microscopy images on the two layers of basement membranes between outer enamel epithelium and the capillaries. The Ronnholm, 1962 paper is by far the only paper not only originally showing the 25-30nm space between the ameloblast distal membrane and the elongating enamel ribbons but also describing this space in detail. We would like to refer our readers to this paper when they would like to know more on this point.

If you pay close attention to these topics, you will find that all the information must be obtained from electron microscopy. The use of electron microscopes was a trend in last century but not now. It is extremely important to bring back a comprehensive set of electron microscope techniques to the field and incorporate genetic and molecular findings with the high-resolution electron microscopic findings to advance our knowledge of tooth development and mineralization. This is what we have been doing for years and will further advance in the future.

We would like to thank you again for your careful evaluation of our manuscript! We made substantial revision to our manuscript based on your and other reviewers’ suggestions. We are very appreciative of your comments!

Round 2

Reviewer 1 Report

Comments and Suggestions for Authors

Dear authors, congratulations! Thanks to my advice, you have further improved your manuscript, but I would ask you to improve your bibliography, as there are some articles that are too old to be used as bibliographical references.

Author Response

Comment 1: Dear authors, congratulations! Thanks to my advice, you have further improved your manuscript, but I would ask you to improve your bibliography, as there are some articles that are too old to be used as bibliographical references.

Response 1: Thank you for your suggestion! When we cited paper, we hope to the cite the most recent and the best ones that provided the best evidence on a certain finding. But certain pieces of information with high quality data are only available in early literature, not recent ones. This is most likely because these old references are good enough to make certain points so that authors of recent papers chose not to unnecessarily repeat them. Therefore, some of the best evidence could come from very old references.

For example, the topic of vascular supply to tooth germs was mostly investigated a long time ago. Only few papers on this topic can be found after 2000, and those were all with different focuses but did not provide sufficient evidence on a very specific finding. The Bernick, 1960 paper is the best one to give a wholistic view of vascular supply in teeth. The Decker, 1963 and 1967 papers provided the clearest and the only electron microscopy images on the two layers of basement membranes between outer enamel epithelium and the capillaries. The Ronnholm, 1962 paper is by far the only paper not only originally showing the 25-30nm space between the ameloblast distal membrane and the elongating enamel ribbons but also describing this space in detail. We would like to refer our readers to this paper when they would like to know more on this point.

If you pay close attention to these topics, you will find that all the information must be obtained from electron microscopy. The use of electron microscopes was a trend in last century but not now. It is extremely important to bring back a comprehensive set of electron microscope techniques to the field and incorporate genetic and molecular findings with the high-resolution electron microscopic findings to advance our knowledge of tooth development and mineralization. This is what we have been doing for years and will further advance in the future.

We would like to thank you again for your careful evaluation of our manuscript!

Round 3

Reviewer 1 Report

Comments and Suggestions for Authors

I ask for no further modification 
Congratulations authors !